# ATR is essential for preservation of cell mechanics and nuclear integrity during interstitial migration

Gururaj Rao Kidiyoor [1], Qingsen Li[1], Giulia Bastianello[1], Christopher Bruhn [1], Irene Giovannetti[1], Adhil Mohamood[1], Galina V. Beznoussenko[1], Alexandre Mironov[1], Matthew Raab[2], Matthieu Piel [2], Umberto Restuccia[1], Vittoria Matafora[1], Angela Bachi [1], Sara Barozzi[1], Dario Parazzoli [1], Emanuela Frittoli[1], Andrea Palamidessi [1], Tito Panciera[3], Stefano Piccolo[1,3], Giorgio Scita [1,4], Paolo Maiuri [1], Kristina M. Havas[1], Zhong-Wei Zhou[5,6], Amit Kumar [7], Jiri Bartek[8,9], Zhao-Qi Wang[5,10] & Marco Foiani [1,4✉]

ATR responds to mechanical stress at the nuclear envelope and mediates envelope-associated repair of aberrant topological DNA states. By combining microscopy, electron microscopic analysis, biophysical and in vivo models, we report that ATR-defective cells exhibit altered nuclear plasticity and YAP delocalization. When subjected to mechanical stress or undergoing interstitial migration, ATR-defective nuclei collapse accumulating nuclear envelope ruptures and perinuclear cGAS, which indicate loss of nuclear envelope integrity, and aberrant perinuclear chromatin status. ATR-defective cells also are defective in neuronal migration during development and in metastatic dissemination from circulating tumor cells. Our findings indicate that ATR ensures mechanical coupling of the cytoskeleton to the nuclear envelope and accompanying regulation of envelope-chromosome association. Thus the repertoire of ATR-regulated biological processes extends well beyond its canonical role in triggering biochemical implementation of the DNA damage response.

[1] IFOM- FIRC Institute of Molecular Oncology, Milan, Italy. [2] Institut Curie/CNRS, Paris, France. [3] University of Padova, Padova, Italy. [4] University of Milan, Milan, Italy. [5] Leibniz Institute on Aging, Fritz Lipmann Institute, Jena, Germany. [6] School of Medicine, Sun Yat-Sen University, Shenzhen, China. [7] Genome and Cell Integrity Lab, CSIR-Indian Institute of Toxicology Research, Lucknow, India. [8] Danish Cancer Society Research Center, Copenhagen, Denmark. [9] Karolinska Institute, Stockholm, Sweden. [10] Friedrich-Schiller University, Jena, Germany. ✉email: marco.foiani@ifom.eu

Mechanical properties of the nucleus and nuclear mechanosensing affects genome integrity, nuclear architecture, gene expression, cell migration, and differentiation[1,2]. The physical properties of the nucleus are conveniently modulated following the inputs from the cell microenvironment or from chromatin dynamics. The nuclear envelope (NE) plays a critical role in this process by connecting the cytoskeleton and the chromatin[1,2].

Ataxia Telangiectasia and Rad3-related protein (ATR) regulates the DNA damage response (DDR)[3] and protects genome integrity by regulating multiple pathways[4]. ATR mutations cause the Seckel syndrome, an autosomal recessive disorder characterized by growth retardation, dwarfism, and microcephaly with mental retardation[5]. We previously reported that ATR directly senses mechanical stress at the NE/chromatin interface and facilitates release of chromatin from the NE[6]. This possibility is supported by the fact that ATR comprises HEAT (huntingtin, elongation factor 3, A subunit of protein phosphatase 2A, and TOR1) repeats, which are elastic connectors, ideal to sense mechanical stimuli[7,8].

Here we explore the possibility that ATR-mediated mechanical communication are also important for the state of the NE itself and, having obtained evidence to this effect, explore its functional implications.

## Results

**ATR is enriched at membranes and actin filaments around the nucleus.** We visualized ATR distribution in exponentially growing HeLa cells by electron microscope (EM) and found ATR in the nucleus, cytosol, and other organelles, including endoplasmic reticulum (ER), Golgi, and mitochondria (Supplementary Fig. 1a). ATR (18.8%) was bound to actin filaments, particularly in the proximity of the NE and more than 20% was bound to cellular membranes (Fig. 1a). Membrane fractionation analysis confirmed that 17% of ATR co-fractionated with membranes, also when nucleic acids were degraded by Benzonase treatment (Supplementary Fig. 1b). We used TopBp1, a chromatin-bound protein, tubulin, a cytoplasmic protein, and Nup133, a NE protein, as controls in our fractionation experiments (Supplementary Fig. 1b). The Kyte and Doolittle[9], and the SOSUI and WoLF PSORT analyses[10], which recognize hydrophobic and membrane-associated domains, respectively, identified seven putative membrane binding and hydrophobic regions in ATR (Supplementary Fig. 1c).

**ATR depletion results in multiple nuclear membrane defects.** Short hairpin RNA (shRNA)-mediated ATR depletion in HeLa cells caused 80% reduction of ATR (Supplementary Fig. 1d) and no obvious cell cycle anomalies (Supplementary Fig. 1e). Immunofluorescence (IF) analysis showed that shATR cells have compromised nuclear morphology, characterized by altered nuclear circularity index, invaginations, and micronuclei (Fig. 1b–e). Similar defects were observed in ATR-depleted U2OS cells (Supplementary Fig. 1f), human ATR Seckel fibroblasts, and non-cycling primary neurons isolated from humanized Seckel mice[11] (Supplementary Fig. 1g, h). ATR$^{flox/-}$ HCT116 cells, which have reduced ATR levels[12] (Supplementary Fig. 1i), also displayed compromised nuclear morphology (Supplementary Fig. 1j, k). We transfected ATR$^{flox/-}$ cells with wild-type green fluorescent protein (GFP) tagged ATR (GFP-ATR) or with a kinase inactive version of GFP-ATR. Although wild-type GFP-ATR rescued the nuclear defects, the mutant form did not (Supplementary Fig. 1l, m). We then performed EM analysis of shATR nuclei (Fig. 1f, Supplementary Fig. 1n–s, and Supplementary Video. 1). ATR-depleted cells exhibited NE invaginations of type II (outer and inner membranes invaginations) and type I (inner membranes

invaginations)[13], associated with condensed chromatin and/or nucleoli (Fig. 1f and Supplementary Fig. 1n–r). NE invaginations also associated with nucleoli forming nucleolar canals that represent intermediates in rRNA export through the NE[14,15] (Fig. 1f and Supplementary Fig. 1r). We also found, within the nucleus, inner membrane invaginations/fragments attached to chromatin and micronuclei (Supplementary Fig. 1r, s).

**ATR depletion alters nuclear mechanical properties.** NE abnormalities can affect the mechanical properties of the nucleus[1,16]. When we measured the elastic modulus of ATR-depleted cells by atomic force microscopy (AFM)[17], we found a reduced elasticity compared to controls (Fig. 2a). As the nucleus is the stiffest organelle in the cell[18], we performed the same analysis on isolated ATR-defective nuclei and found, again, a reduced elasticity, compared to controls (Fig. 2b). Acute treatment with ATR inhibitors for 4 h did not alter nuclear stiffness (Supplementary Fig. 2a). Hence, the reduced nuclear elasticity results from chronic ATR depletion.

**Lipid composition of the NE is altered in ATR-defective cells.** Nuclear stiffness is influenced by lamins and nuclear membrane fluidity. We did not find significant alterations in Lamins protein levels or in their relative cellular localization in ATR-depleted cells (Supplementary Fig. 2b). However, when we measured the lipid composition of isolated nuclear membranes from ATR-depleted and control cells, out of 855 lipid species analyzed, we observed significant differences in the phosphatidylcholine (PC)/phosphatidylethanolamine (PE) ratio (Fig. 2c). In particular, we observed a specific altered ratio in the 18-carbon and 17-carbon lipid species that represent the most common lipids in membranes (Fig. 2c). Of note, when we performed a whole cell metabolic analysis, we did not observe specific alterations at the levels of PC/PE ratios.

**ATR depletion alters chromatin organization.** Chromatin conformation and distribution can also influence mechanical properties of the nucleus[19]. We first performed the DNAse I sensitivity assay[20] to analyze the chromatin state of control and ATR-depleted cells (Supplementary Fig. 2c). DNAse preferentially cleaves euchromatin, which is more accessible than heterochromatin to its enzymatic activity[21]. We found that at early time points, ATR-defective cells exhibited a higher level of undigested DNA that failed to migrate in the gel and likely resulted from heterochromatin accumulation (Supplementary Fig. 2c). Perinuclear chromatin is generally in the heterochromatic state and is influenced by the levels of H3K9 trimethylation[22]. ATR-defective cells exhibited a 20% increase in K9 trimethylated histone H3 compared to control (Supplementary Fig. 2d), implying that ATR depletion promotes increased heterochromatization. This conclusion is confirmed by the fluorescence energy transfer (FRET)-based fluorescent lifetime imaging microscopy (FLIM) assay utilizing GFP- and mCherry-tagged histone H2B to measure chromatin compaction[23]. We found that, under unperturbed conditions, loss of ATR causes a reduction in the fluorescent lifetime signal, indicating that H2B histones are more compacted (Fig. 2d). However, we did not observe changes at the level of FRET signal or H3K9 trimethylation upon short-term inhibition of ATR using kinase inhibitors (Fig. 2d and Supplementary Fig. 2e), suggesting that the aberrant chromatin state owing to ATR depletion results from long-term effects.

**ATR depletion affects LINC-mediated nuclear-cytoskeleton connections.** Another parameter affecting the mechanical properties of the nucleus is the connection between the NE and the cytoskeleton, which is mediated by the Linker of Nucleoskeleton and Cytoskeleton (LINC) complex[24]. We measured this parameter in cells with or without ATR, using a FRET sensor of Nesprin 2G[25],

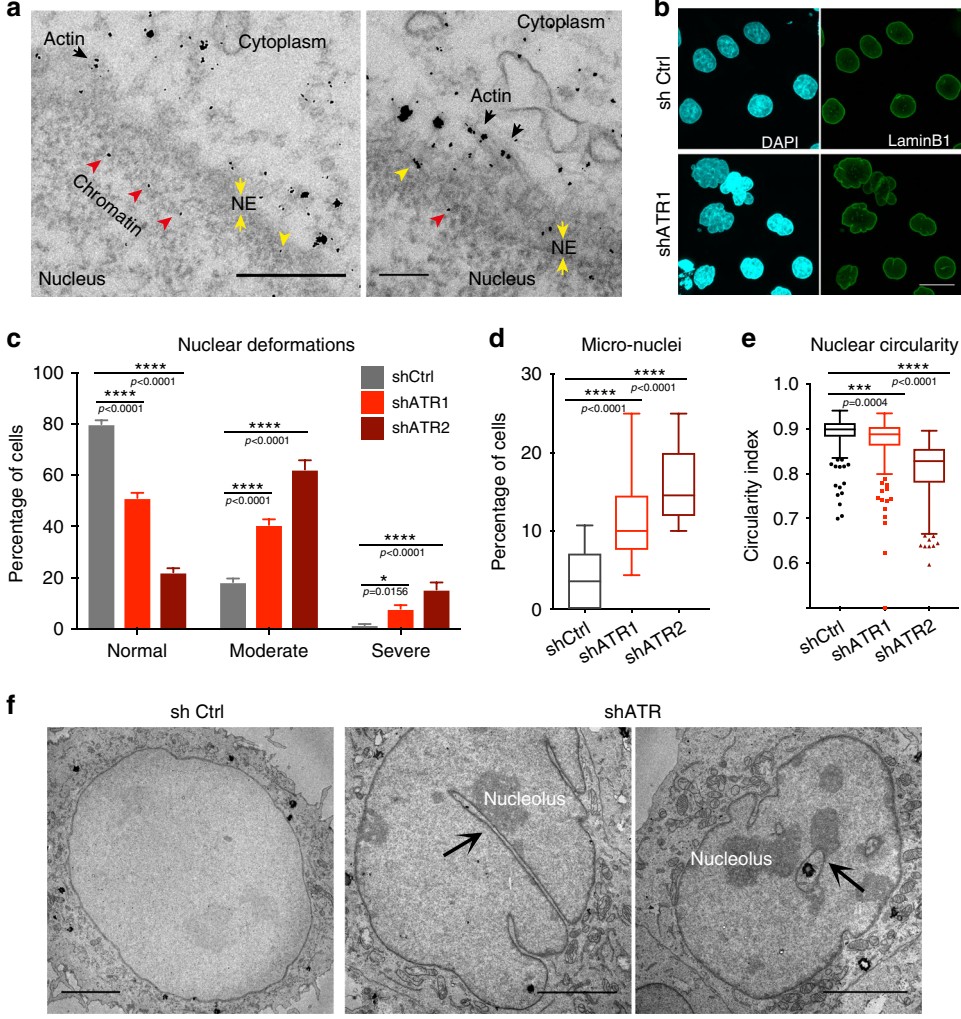

**Fig. 1 ATR interacts with membranes and preserves nuclear morphology. a** Images from routine 60 nm EM section with enrichment of nano-gold-labeled ATR in the perinuclear region. Colored arrowheads indicate actin-associated (black), chromatin-associated (red), and membrane-associated ATR (yellow), respectively (scale bar 500 ηm for the right and 200 ηm for the left panel). **b** Defective nuclear morphology of ATR-depleted HeLa cells visualized with immunofluorescence of Lamin B1 (NE) and DAPI (DNA) (scale bar = 50 μm). **c** Quantification of nuclear deformations by manual sorting based on their degree of deformation: Normal, mildly deformed and severely deformed (n = 515, 525, 229 cells for shCtrl, shATR1, and ShATR2, respectively). Quantifications of **d** micronuclei (n = 515, 525, and 229 cells for shCtrl, shATR1, and ShATR2, respectively) and **e** nuclear circularity index (n = 288, 253, and 215 cells for shCtrl, shATR1, and ShATR2, respectively; N = 3 independent experiments). **f** EM images of the nucleus from control and ATR-depleted HeLa cells. Arrowheads indicate invaginations with nucleoli attachment (scale bar 5 μm). Bar graphs presented as mean ± SEM and box plot whiskers and outliers plotted with Tukey's method. p-values calculated using two-way ANOVA test for **c** or for **d**. **e** Ordinary one-way ANOVA test with Dunnett's multiple comparisons test (****P < 0.0001, ***P < 0.001, **P < 0.01, *P < 0.05; NS, not significant). Source Data file contains source data and all additional details of statistical analysis.

a component of the LINC complex[24]. This assay measures relative changes in the molecular length between the actin-binding domain and the membrane-binding domain of Nesprin 2G. A low FRET signal means that the two domains are far apart, implying that Nesprin 2 is connected to the cytoskeleton and the NE. Cells lacking functional ATR, either through acute inhibition or following long-term depletion, exhibited an increased FRET signal at the NE (Fig. 2e). This result indicates that the actin-binding and the membrane-binding domains of the linker molecule are in close proximity, presumably due to an altered NE-cytoskeleton connection. This finding is also consistent with Nesprin 2 influencing the NE tension, thus suggesting that ATR defects may also cause a reduced NE tension.

**ATR depletion causes accumulation of YAP in the cytoplasm.** The LINC-mediated mechanical coupling between the NE and the

cytoskeleton also influences the nuclear accumulation of YAP, a key mechanosensing transcriptional activator[26,27]. We investigated whether ATR depletion affected YAP nuclear accumulation (Fig. 2f). We found that, although in control cells YAP distribution was mostly nuclear, the YAP cytoplasmic fraction increased significantly in ATR-defective cells (Fig. 2f). Moreover, following ATR depletion, cells accumulated the cytoplasmic form of YAP, phosphorylated in Serine 127 (Fig. 2g). To exclude the possibility that the change in YAP localization could be a nonspecific consequence of YAP leakage due to NE disruption, we tested whether interfering with Exportin1 (with the Leptomycin-B specific inhibitor), essential for YAP nuclear exit under physiological conditions, could counteract the effect of ATR depletion. Treatment of ATR-depleted cells with Leptomycin-B rescued nuclear YAP levels, with concomitant decrease of phosphorylated YAP (Supplementary Fig. 2f, g). Following an acute treatment with an ATR

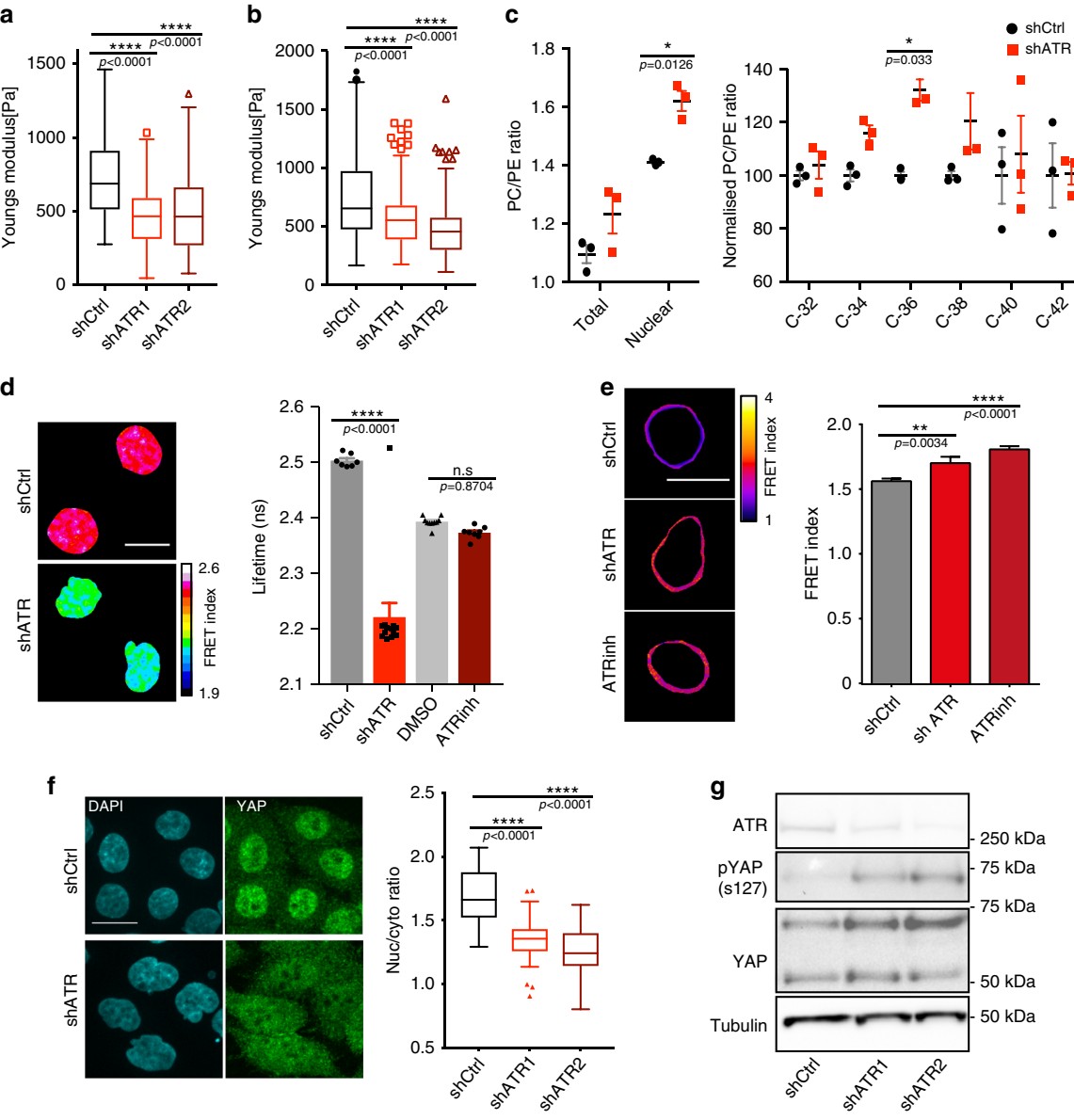

**Fig. 2 ATR preserves nuclear mechanics. a**, **b** Elastic modulus measurements using AFM. **a** Cellular stiffness (n = 171, 161, and 179 measurements for shCtrl, shATR1, and ShATR2, respectively) and **b** stiffness of isolated nuclei from control and ATR-depleted cells (n = 144, 110, and 98 measurements; N = 2 independent experiments). **c** Membrane phospholipid composition analysis; total cell and nuclear membrane PC/PE ratio, nuclear PC/PE ratio of individual species (from three biological and two technical replicates; two-way ANOVA test; Bonferroni's multiple comparisons test; N = 2 independent experiments). **d** FLIM-FRET analysis of cells expressing H2B-GFP and H2B-mCherry; sample images of Fluorescent lifetime (FLIM-FRET index) and overall Lifetime in Hela cells infected with shATR (n = 13) or control (n = 7), or treated with DMSO (n = 10) or ATR inhibitor for 4 h (n = 8). **e** Examples of FRET signals at the NE as measured by Nesprin-2 FRET sensor in control, shATR, and control cells in the presence of ATR inhibitor (3 h), and quantification of FRET signals (n = 112, 88, and 96 measurements for shCtrl, shATR1, and ShATR2 respectively; data pooled from two or three independent experiments). **f** Immunofluorescent images of YAP cellular distribution in control and shATR cells, (below) quantifications of nuclear to cytoplasmic YAP signal ratio (n = 69, 66, and 58 shCtrl, shATR1, and ShATR2, respectively). **g** Western blotting of ser127-phosphorylated YAP (N = 2 independent experiments. Uncropped images available in Source Data file). Scale bar is 20 μm in all images. Bar graphs presented as mean ± SEM and box plot whiskers and outliers plotted using Tukey's method in prism7 software. p-values calculated using one-way ANOVA test with Tukey's multiple comparisons test (for **d**) or Dunnett's multiple comparisons test (for **a**, **b**, **e**, **f**). (****P < 0.0001; ***P < 0.001; **P < 0.01; *P < 0.05; n.s., not significant).

inhibitor for 3 h, again, we found enrichment of YAP in the cytoplasm and this effect was abolished by Leptomycin-B treatment (Supplementary Fig. 2h, i).

**ATR-defective nuclei collapse following mechanical compression.** The previous results suggest that ATR is required for the normal response of the nucleus to mechanical forces and, as the responses of the cell are dominated by effects on the nucleus, that ATR may also influence cell mechanoresponsiveness. We therefore analyzed the response of ATR-defective cells to mechanical stress using a microfluidic device able to compress cells with a linear step-wise increment of mechanical forces (see "Methods") (Fig. 3a). GFP-tagged cGAS, a cytoplasmic DNA-binding protein, was used as an NE damage marker[28]. We compressed cells expressing cGAS-GFP using the microfluidic device by applying

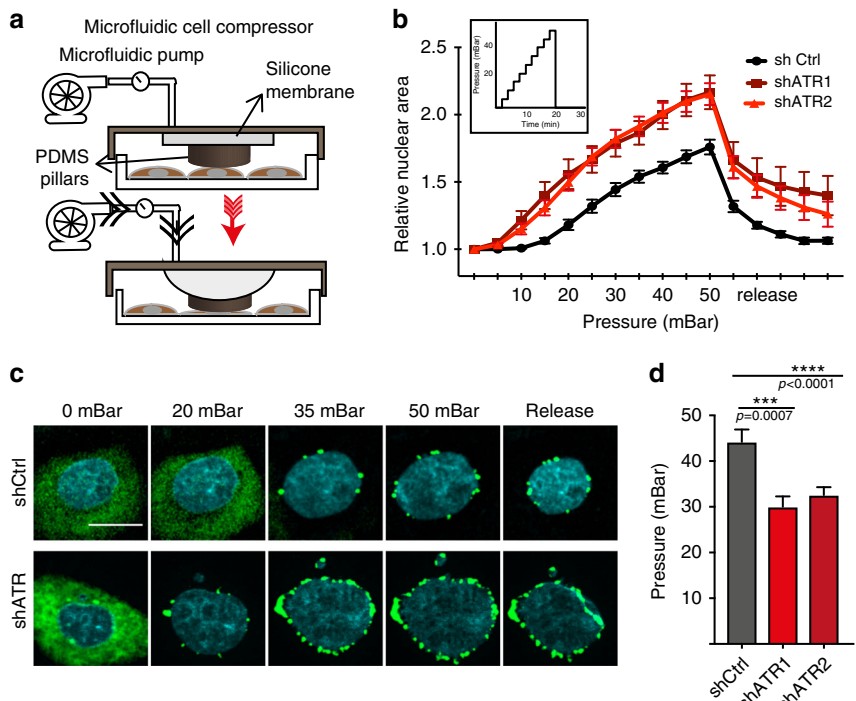

**Fig. 3 ATR-defective cells undergo nuclear collapse upon mechanical compression.** Mechanical compression using a microfluidic compression device. **a** Design and **b** graph of compression-induced nuclear deformation with respect to the pressure applied, represented as ratio of its initial uncompressed area. In inlet, graph representing step-wise increase of pressure used for the experiment. **c** Time-lapse images of perinuclear cGAS foci formation in control and ATR-depleted cells under compression. **d** Amount of pressure applied through the microfluidic pump during the time of NE rupture (formation of first foci of cGAS) ($n = 19$, 24, and 21 shCtrl, shATR1, and ShATR2, respectively; data pooled from $N = 2$ independent replicates). All graphs presented as mean ± SEM. $p$-values are calculated using two-way ANOVA test and Tukey's multiple comparisons test for **b** and one-way ANOVA test with Dunnett's multiple comparisons test for **d** (**** if $P < 0.0001$, *** if $P < 0.001$).

ten steps of compression with a range of pressure spanning 0 to 50 mBar, with an increment of 5 mBar at each step. ATR-depleted cells accumulated perinuclear cGAS-GFP foci in the range between 15 and 30 mBar, whereas in control cells cGAS-GFP foci started to appear only at 45 mBar (Fig. 3b–d). We also analyzed the recovery from cell compression and found that control cells restored the initial nuclear size following de-compression, but ATR-defective nuclei did not completely revert back to the original size (Fig. 3b, c). We conclude that ATR-defective cells, experiencing compression forces, undergo irreversible nuclear deformation and collapse followed by NE ruptures, thus exposing DNA into the cytoplasm, and accumulating perinuclear cGAS.

**ATR defects impair interstitial migration.** When migrating through tight spaces, nuclei experience tremendous mechanical stress that causes chromatin compaction and nuclear ruptures. Cell survival under these conditions depends on cellular pathways controlling nuclear integrity and nuclear membrane repair[28–30]. The previous sections suggest that ATR defects should compromise the ability of cells to migrate through narrow pores. To explore this possibility, we analyzed the contribution of ATR on interstitial migration using in vitro experiments. Moreover, as nuclear stress and chromatin compaction are prominent features during neurogenesis and metastasis[2,31,32], we also addressed whether ATR influences neurogenesis and metastasis by in vivo experiments.

We analyzed H2B-mCherry-expressing shATR and control HeLa cells migrating through 4 µm-wide and 15 µm-long microfabricated constrictions[28] (Fig. 4a). The fraction of cells undergoing nuclear collapse and cell death (Fig. 4a, see also Supplementary Movie. 2) while engaging the constrictions were increased in the

absence of ATR (Fig. 4b). Similar results were observed in cells treated with ATR inhibitors (Supplementary Fig. 3a). We measured DNA damage occurrence by counting 53BP1 foci of cells in the constrictions and found comparable level of foci numbers between normal and ATR-defective cells (Fig. 4c). Given the well-established links between ATR activity and cell cycle checkpoints, we tested whether the migration defects of ATR-depleted cells were also connected to the cell cycle. Using U2OS-FUCCI cells that mark different phases of cell cycle[33], we found that cell death in shATR cells was not dependent on the cell cycle phase (Supplementary Fig. 3b). We found comparable range of 53BP1-GFP foci in ATR-inhibited cells and in control cells; moreover, the presence of 53BP1 foci did not correlate with cell death (Fig. 4c). This finding is in agreement with previous reports[28,29] showing that 53BP1 foci accumulate in a Ataxia-Telangiectasia mutated protein (ATM)-dependent manner. We note that ATM is fully functional in ATR-defective cells. Therefore, nuclear collapse and cell death of ATR-defective cells in constrictions does not correlate with increased DNA damage or cell cycle stage. Using the Nesprin 2G FRET sensor, we found lower FRET intensity at the leading edge of the nucleus and higher intensity at the lagging edge (Fig. 4d, e), suggesting that cells respond to migration-induced mechanical stress by modulating the cytoskeleton-NE connection at the leading edge of the NE. ATR inhibition caused a general increase in FRET signal (Supplementary Fig. 3c, d) and abolished the differential coupling of the cytoskeleton-NE connection in the nuclei engaged in the constrictions (Fig. 4e). As a control, we used a headless form of Nesprin 2G sensor, which cannot bind cytoskeleton, and found that it did not exhibit significant changes in FRET signals in control or ATR-defective cells (Supplementary Fig. 3e).

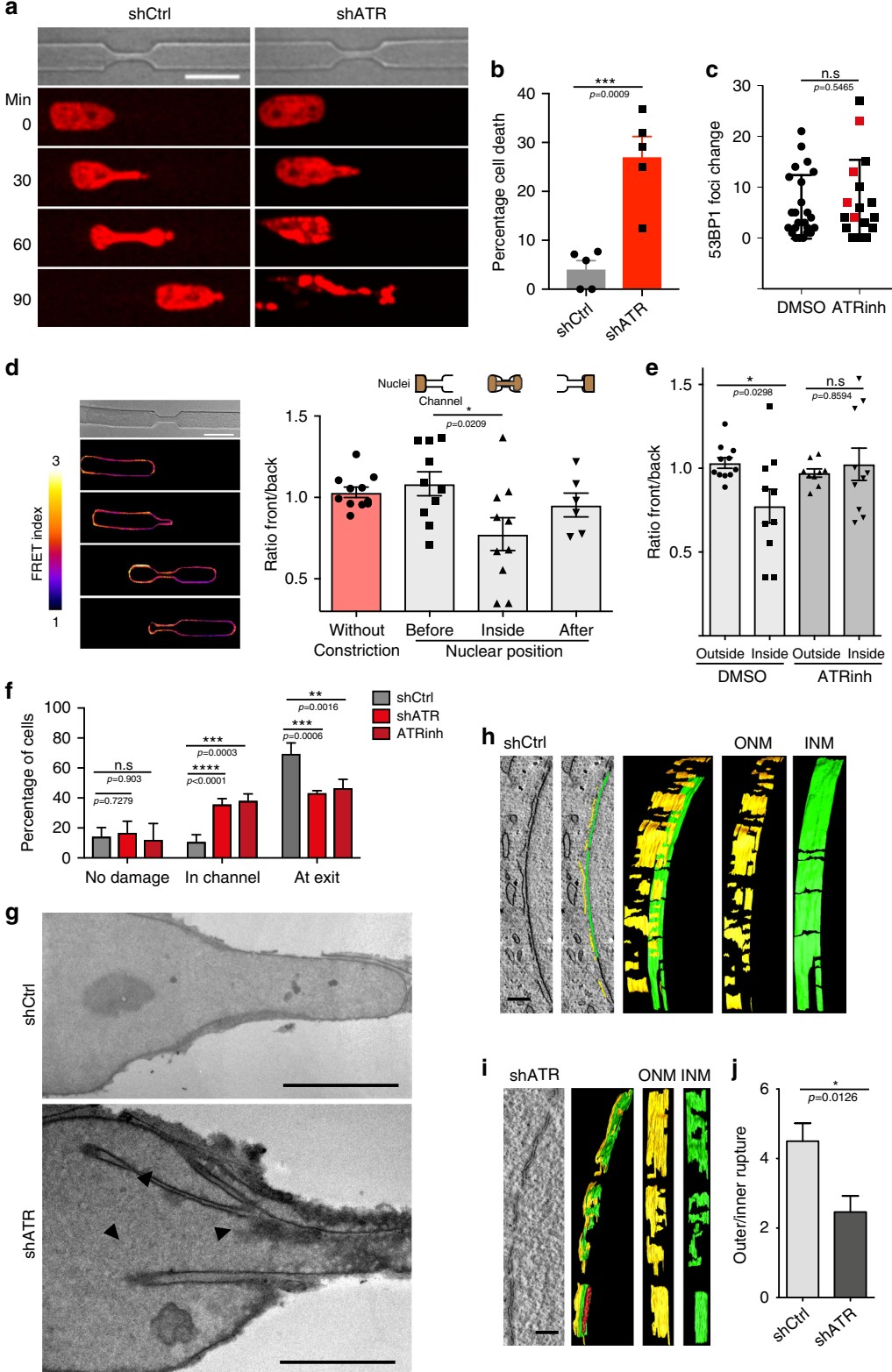

We then analyzed the cGAS-GFP foci distribution during interstitial migration (Fig. 4f). We found that, compared to normal cells, in ATR-defective cells cGAS-GFP foci appeared much earlier at the leading tip of nuclei engaged in the constrictions, suggesting that ATR defects enhance NE fragility during interstitial migration. To directly analyze the integrity of the NE experiencing interstitial migration, we performed EM analysis of cells migrating across constrictions with or without ATR (Fig. 4g and Supplementary Fig. 3f–i). Control cells approached the constrictions by deforming their nuclei at the leading edge without aberrant NE structures (Fig. 4g and Supplementary Fig. 3f) or, occasionally, with well-organized NE invaginations parallel to the direction of migration (Supplementary Fig. 3g). Conversely, most of ATR-depleted nuclei were deformed, exhibiting several NE invaginations in the front

**Fig. 4 ATR-defective nuclei are inefficient in migrating through narrow pores. a** Snapshots of H2B-mCherry labeled control and shATR nuclei passing through constriction. **b** Cell death measured as the percentage of engaged cells that burst at the constriction ($n = 122$ and 88 cells for shCtrl and shATR1; data pooled from three independent experiments). **c** Quantification of 53BP1-GFP foci generated due to constriction in HeLa cells expressing 53BP1-GFP in the presence of DMSO or ATR inhibitor, VE-821 ($n = 24$ and 17 for DMSO and ATRinh; pooled from two independent experiments). Cells that undergo cell death in the constriction are highlighted in red (for ATRinh). **d, e** FRET signal measurements of cells engaged in constrictions. **d** Images of FRET signal at various stages of migration through the constriction and measurement of signal ratio between front (leading half of the nucleus) and back (lagging half of the nucleus) of a nuclei at various stages of migration ($n = 11$, 10, 10, and 6, respectively). **e** Ratio of front to back FRET signal in migrating cells (inside or outside the constriction) in the presence DMSO or ATRinh ($n = 11$, 10, 9, and 10; data from 2 to 3 experiments). **f** Quantification of nuclear position in the constriction during the first cGAS foci formation ($n = 47$, 43, and 28; numbers pooled from 3 experiments). **g** EM images of control and shATR nuclei in constriction (routine 200 nm EM sections). Arrowheads indicate invaginations and NE attached chromatin or nucleoli. **h** 3D reconstruction of NE at the leading edge from control nucleus in constriction. Green color indicates inner nuclear membrane (INM) and yellow indicates outer nuclear membrane (ONM). **i** 3D reconstruction of NE section from leading edge of shATR nucleus in constriction. **j** Quantification of ratio between number of inner nuclear membrane breaks to that of the outer membrane ($n = 15$ and 13). Scale bar for **a, d** is 20 μm, for **g** is 9 μm, and for **i, h** is 200 ηm). Bar graphs presented as mean ± SEM and dot-plot as mean ± SD. $P$-value calculated using two-tailed Student's $t$-test for **b, c, j**. One-way ANOVA for **d, e** with Tukey's or Sidak's multiple comparisons test, and two-way ANOVA for **f** (****$P < 0.0001$, ***$P < 0.001$, **$P < 0.01$, and *$P < 0.05$; n.s., not significant).

part of the nucleus, often associated with semicondensed chromatin or nucleoli (Fig. 4g and Supplementary Fig. 3f, h). Both control and ATR-depleted cells exhibited sporadic NE ruptures (Fig. 4h). Our EM analysis allowed us to establish that control cells accumulated extensive outer membrane ruptures and fewer inner membrane ruptures at the leading edge of the nucleus (Fig. 4h), whereas ATR-depleted cells accumulated extensive NE ruptures involving both the outer and inner membranes, with a higher frequency of inner membrane damage (Fig. 4i, j). Moreover, we noticed that the leading edge of ATR-defective nuclei exhibited NE portions with a disorganized distribution of outer and inner membranes, likely due to aberrant nuclear membrane remodeling. The NE at the rear part of the nucleus was normal and comparable in control and in ATR-depleted cells (Supplementary Fig. 3i). Hence, as soon as ATR-defective cells engage narrow constrictions, they fail to adequately respond to the mechanical stress arising at the leading edge of the nucleus and undergo NE deformation and extensive NE damage, which, in turn, cause nuclear collapse and cell death. We failed to observe NE ruptures in ATR-defective cells grown under normal conditions, suggesting that they represent a consequence of mechanical compression. Although the intrinsic alterations of the mechanical properties of ATR-defective nuclei may not affect cell viability under normal conditions, the consequences of nuclear collapse following mechanical stress certainly contribute to cell lethality when cells are forced through narrow passages.

**ATR influences neurogenesis and metastatic dissemination.** During neurogenesis and metastasis, cells migrate through narrow places. The in vitro observations described above suggest that ATR may play a relevant role in these processes. We analyzed the contribution of ATR in neurogenesis by performing a transwell migration assay of neuroprogenitors isolated from ATR-conditional knockout mouse brain (E13.5 days) (Fig. 5a). ATR depletion impaired migration of neurosphere-derived cells through 3 or 8 μm pore size membranes and, as expected, the defect was more pronounced in the smaller pore size. We next depleted ATR in vivo, in a developing mouse brain. GFP-tagged shRNAs against ATR were electroporated into a developing brain (at day 14.5), to selectively deplete ATR in a subpopulation of migrating neurons (Fig. 5b). Cortical plates from these brains were analyzed in later stages (E18.5) of embryonic development. By using two independent shRNAs against ATR, we observed a compromised neuronal migration in the cortical plate: although many of the neurons transfected with Luciferase (control) reached the top layers of cortical plate, ATR-depleted neurons were stuck in lower layers (Fig. 5b, c).

We then examined the contribution of ATR in the migration of cancer cells. We injected equal number of shRNA control and ATR-depleted HeLa cells labeled with a vital dye into the tail vein of immunocompromised mice and recovered lung disseminated cells at 2 and 48 h after injection (see scheme in Fig. 5d). We found a significant reduction of fluorescent-positive shATR cells in the lung 48 h after injection compared to controls (Fig. 5e, f), indicating that ATR is essential to allow cells to sustain the harsh mechanical environment imposed by blood flow and extravasation.

**ATR interactors known to influence nuclear mechano-response.** To identify potential ATR interactors and targets contributing to nuclear mechanics and dynamics, we performed high-resolution mass spectrometry screens (IP-liquid chromatography (LC)-MS/MS) in exponentially growing U2OS cells expressing GFP-ATR. Combining data from three SILAC (stable isotopes labeling with amino acids in cell culture) quantitative approaches (Supplementary Fig. 4a), we obtained 479 unique interactors of ATR (analysis details in "Methods" and Supplementary Data 1). Our ATR interactome exhibited a 55% enrichment (265/479) for proteins with (S/T-Q) motif, a potential targets of ATR phosphorylation, while such proteins represent only 22% (5137/17522) in the total phospho-proteins of the PhosphoSitePlus database[34] (Supplementary Fig. 4b). Several of these interactors were reported to be phosphorylated during S phase, mitosis[35] (71/479) and in response to DNA damage when ATR is hyperactive[4] (48/479) (Supplementary Fig. 4c).

We found several ATR interactors for which previous studies have identified roles in the mechanical properties of the nucleus and whose depletion mimics, at least in part, some of the phenotypes observed in ATR-defective cells (Fig. 6a and Supplementary Fig. 4d–f). TOPII, an ATR/ATM phosphorylation target[4], is involved in modulating DNA topology in S phase and in prophase to deal with the mechanical stress caused by chromosome dynamics. Moreover, genetic evidence[36] suggests that Top2 activity is restrained by Mec1[ATR]. We also found HDAC2 and CHD4, which are members of the Nuclear Remodeling and Deacetylating (NRD) complex, previously identified as ATR interactors[37]. An ATR-mediated regulation of the TOP2 and NRD complex could account for the phenotypes associated with the heterochromatic and condensed chromatin observed in ATR-defective cells. The screen identified four proteins of the nuclear pore complex (Nup50, 107, 133, 160), regulating nuclear transport, centrosome attachment to the NE during mitosis, as well as YAP mechanotransduction[26]. In addition, we identified several transport proteins including Exportin1 (XPO1/CRM1), an ATR/ATM phosphorylation target[4], involved in rRNA transport[38], which might be connected to the accumulation of nucleolar canals described in this study. We also

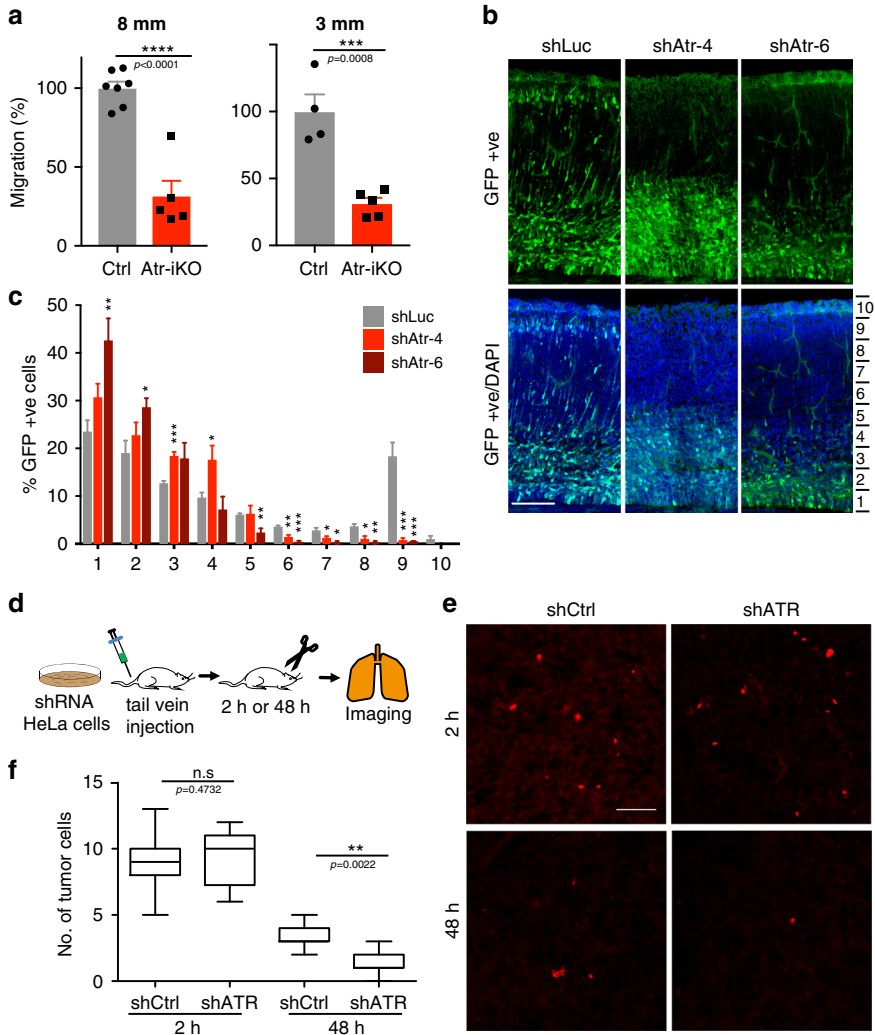

**Fig. 5 Loss of ATR dampens neuronal migration and tumor cell circulation. a** Relative migration of neuroprogenitors isolated from Atr-iKO mice brain were plated into 8 μm ($n = 7$, 5 cell lines) or 3 μm ($n = 4$, 5 cell lines) polycarbonate membrane insert (ThinCert™), allowed to migrate for 20 h, then fixed and counted (neuroprogenitors isolated from 13 embryos from 2 pregnant mice independently). **b, c** In vivo neuronal migration: **b** images of cortical plates from E18.5 embryos. **c** Percentage of GFP-positive (GFP+) cells present in different equally divided segments of E18.5 brain cortex ($n = 4$ animals, 5 section, 2270 GFP+ cells for shLuc; $n = 4$ animals, 5 section, 2475 GFP+ cells for shAtr-4; $n = 2$ animals, 3 section, 730 GFP+ cells for shAtr-6. Statistical comparisons performed between shLuc individually with shAtr-4 or shAtr-6 from the same layer). **d** Scheme of in vivo homing assay. Control and shATR HeLa cells labeled with vital dye were injected into the tail vein of immune-compromised mice. Sections of the lung were collected after 2 and 48 h, respectively. **e** Images of lung surface with labeled HeLa cells residing on them. **f** Quantifications of cells/field at 2 and 48 h, respectively ($n = 25$ images from 5 mice and 15 images from 3 mice (for 2 and 48 h) for control; 20 images from 4 mice (for 2 h) and 25 images from 5 mice (for 48 h) for shATR). Scale bar is 100 μm in all images. Bar graphs presented as mean ± SEM and box plot whiskers and outliers plotted using Tukey's method in prism-7 software. $p$-values calculated using two-tailed Student's $t$-test or one-way ANOVA test with Tukey's multiple comparisons test (****$P < 0.0001$, ***$P < 0.001$, **$P < 0.01$, *$P < 0.05$, n.s., not significant).

identified Nesprin-2. Intriguingly, Nesprin-2-defective cells also exhibit NE invaginations, and chromatin architecture and nuclear mechanics dysfunctions[39,40]. Moreover, loss of Nesprin-2 leads to defective neuronal migration in developing mice brain[41]. We confirmed the ATR–Nesprin-2 interaction by immunoprecipitation (IP) followed by western blotting (Fig. 6b and Supplementary Fig. 4h) and proximity ligation assay (PLA) (Fig. 6c). PLA showed that the number of ATR–Nesprin-2 foci at the NE increased in cells undergoing chromatin condensation in prophase (Fig. 6d and controls in Supplementary Fig. 4i). This observation, combined with the previous result showing that ATR influences nesprin 2 function (Fig. 2e), suggests that ATR and Nesprin-2 dynamically interact at the NE in response to mechanical stress. ATR depletion did not affect Nesprin-2, protein levels, or intracellular localization

(Supplementary Fig. 4g). In sum, the ATR-phosphointeractome reveals a number of ATR targets involved in the response of the NE to mechanical strains. Although certain targets might be directly regulated by ATR to allow cells to properly respond to mechanical stress, a set of ATR interactors might mediate the recruitment of ATR to the NE or even promote ATR activation when nuclei are stressed. Of note, we did not find ATR interactors involved in lipid metabolism.

## Discussion

Our findings unravel a non-canonical role for ATR in maintaining the normal properties of the NE and in mediating a communication between the cytoplasm and the nuclear interior (chromatin/chromosomes). These observations, together with previous

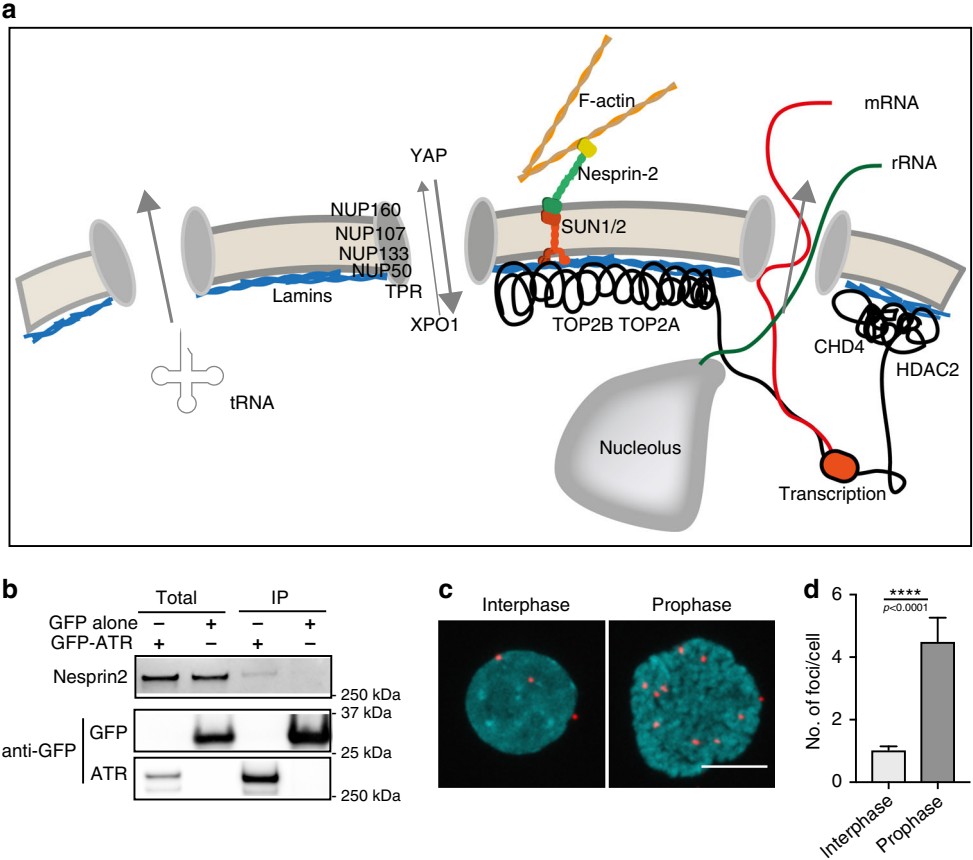

**Fig. 6 ATR interactors. a** Schematic summary of ATR interactors and their relative roles at the nuclear envelope and perinuclear areas **b** Validation of Nesprin-2 and ATR interaction using GFP-ATR immunoprecipitation and western blot analysis (full blots of Nesprin-2 with molecular weights included in Supplementary Fig. 4h and in Source data file). **c** Example of ATR–Nesprin-2 interaction foci generated by proximity ligation assay (PLA) in interphase and prophase cells (scale bar = 10 μm). **d** Quantification of total number of foci present in interphase and prophase cells ($n = 65$ for interphase and 10 for mitosis, pooled from 2 independent experiments). Graph presented as mean ± SEM. $p$-value calculated using two-tailed Student's $t$-test.

findings showing that (i) ATR is a giant HEAT repeat protein, therefore ideal for sensing mechanical stress[7,8]; (ii) ATR reliefs the mechanical stress at the NE exerted by the mRNA export machinery[42]; and (iii) ATR relocalizes at the NE following mechanical stress[6,43], suggest that ATR might directly sense and transduce mechanical stress in general and at the NE.

We described a variety of nuclear defects owing to ATR depletion or inactivation, some of which are evident already in unchallenged cells; others become obvious when cells (and thus their nuclei, which are their primary load-bearing element) experience mechanical deformations. Although some of the nuclear defects likely represent a direct consequence of ATR inactivation/depletion, others may be related to long-term adaptive responses to limiting ATR. We also identified a set of ATR interactors that might be involved in the cellular response to mechanical stress by mediating ATR recruitment at the NE, transducing the mechanical stress signals or in triggering short and long-term response pathways. Finally, we show that ATR is essential for neurogenesis and cancer cell migration, two processes in which cells and their nuclei must squeeze through tight spaces.

**Nuclear defects in unchallenged ATR-depleted cells.** Under normal conditions, ATR-defective cells exhibit NE invaginations tethered with semicondensed chromatin and nucleoli. These nuclear defects appear only upon long-term depletion of ATR. Considering that (i) condensation of perinuclear chromatin and nucleolar canals generates transient NE invaginations[14,44]; (ii) ATR-defective cells are unable to coordinate NEBD with chromatin

condensation, as they exhibit a slow condensation process[6]; and (iii) ATR has been involved in coordinating RNA export through the NE with topological stress[45], the most likely possibility is that in ATR-defective cells NE remains deformed due to the inability to efficiently separate from condensed chromatin and to complete the export of nucleolar RNA species.

NE invaginations are rare in normal cells, as the NE efficiently recovers its original shape. This process requires the separation of condensed and transcribed chromatin from the NE, and is influenced by NE remodeling activities and by proteins modulating the topological and epigenetic context of chromatin and nuclear transport. ATR regulates type II A and B topoisomerases and condensins[4], as well as the NRD complex[37] that controls chromatin epigenetics. It is possible that, in ATR-defective cells, deregulated topological and condensation activities may cause nuclear membranes invaginations and ruptures, nuclear fragmentation, and micronuclei formation; the heterochromatic and heavily condensed chromatin could instead result from the deregulation of the NRD complex[46]. Recent observations[47] showed that H3K9 trimethylation marked heterochromatin levels rearrange in response to mechanical stress at the NE and the recovery of the nucleus. In this scenario, our finding that a long-term depletion of ATR accumulates hypercompacted chromatin at the nuclear periphery and elevated levels of H3K9 trimethylation may therefore reflect the inability of ATR-depleted cells to properly recover from nuclear stress. The hypercompacted chromatin at the nuclear periphery and the consequent reduction in nucleosome packaging density in the rest of the nucleoplasm

might contribute to explain the low nuclear stiffness of ATR-defective cells.

**Nuclear membrane defects owing to ATR depletion**. The lack of coordination between chromatin condensation and NEBD during cell division in ATR-defective cells causes accumulation of semi-condensed chromatin attached to NE fragments[6]. Moreover, our EM analysis showed that ATR-depleted cells accumulate membrane ruptures already under unperturbed conditions and, following nuclear deformation during interstitial migration, they exhibit massive breakage of the outer nuclear membrane and aberrant membrane remodeling. The ESCRTIII complex plays a key role in sealing membrane holes in the reforming NE during mitotic exit[48] and in repairing the NE upon migration-induced rupture[28,29]. It is possible that the activity of the ESCRTIII complex becomes limiting in ATR-defective cells, due to the massive damage of nuclear membranes. The extensive NE damage and remodeling in ATR-defective cells may also represent the primary cause of the aberrant phospholipid composition of their nuclear membranes. In agreement with this hypothesis, the aberrant phospholipid composition of the NE in ATR-defective cells does not reflect a direct metabolic problem and we failed to identify ATR interactors involved in lipid metabolism.

**Altered NE-cytoskeleton coupling upon ATR inactivation**. A key finding of this work is that ATR is a component of the cell mechanotransduction machinery by ensuring appropriate mechanical coupling of the cytoskeleton to the NE. The NE is exposed to forces acting in opposite directions: forces deriving from chromatin dynamics (as outlined above) and opposite forces generated by extracellular matrix (ECM) attachment and conveyed to the NE through the LINC complex and the NE-associated cytoskeleton. ATR orchestrates the integration of all these mechanical inputs, by regulating at once NE dynamics, chromatin condensation, and the LINC function as the mechanosensory properties of the entire cell; this is visualized by the here discovered ATR-YAP mechano-signaling axis.

Although abnormalities in nuclear shape and mechanics can impact on genome integrity by generating chromatin fragmentation[36] and fork collapse[42], under normal conditions, the nuclei remain relatively stable as well as the NE. However, at raising levels of mechanical strain, cells must promptly respond to mechanical stress. Here we show that ATR is critical also for the nuclear response to more severe mechanical challenges. ATR-defective cells fail to properly respond to sub-lethal compression forces and undergo extensive nuclear collapse characterized by NE ruptures. In turn, this imposes an additional stress at the level of nuclear membrane remodeling, as revealed by the presence of mixed portions of the outer and inner membranes at the NE engaged in the constrictions. NE fragmentation under high level of mechanical stress exposes nuclear DNA into the cytoplasm, leading to activation of the cGAS-STING pathway[49]. The functional consequences of the ATR and cGAS connection remain unexplored but may hold relevant pathological implications, particularly in tissues undergoing mechanical stress.

**Pathological consequences owing to ATR defects**. Our observations indicate that, when the nucleus engages the narrow constrictions, the LINC complexes at the leading edge of the nucleus are tightly bound to the cytoskeleton and the mechanical strain generates extensive ruptures at the outer nuclear membranes. Hence, the nucleus during interstitial migration is polarized at the level of NE. ATR-defective cells fail to maintain the coupling between Nesprin-2 and the cytoskeleton, and to polarize the NE under these conditions and accumulate NE

invaginations and extensive ruptures at both nuclear membranes at the leading edge of the nucleus. Moreover, the extensive NE invaginations tethered with semicondensed chromatin may hinder efficient nuclear squeezing and prevent an efficient repair of the nuclear membranes.

Cancer cell migration through the ECM requires nuclear deformability, particularly when cells must meander through dense and highly crosslinked collagen type I-rich stroma, extravagate, and sustain the harsh conditions of the blood circulation before extravasating, passing through pores as small as 2 μm in diameter[50]. In fact, altered NE morphology is typical of cancer cells and crucial in the tumor grade assessment, and correlates with prognosis. Cancer cells can adapt to metastatic migration by deregulating the expression of Lamins, but a certain degree of NE stiffness is required to allow a productive migration and to prevent massive NE ruptures[50]. Our observations suggest that ATR activity might be therefore beneficial for cancer cell migration, thus implying that ATR might play opposite roles in cancer progression, by preventing genome instability and by promoting metastasis. Along this idea, it is interesting to note that our experiments indicate that ATR depletion impairs three-dimensional (3D) invasion and lung homing of cancer cells.

Our findings describe a variety of nuclear defects and their pathological consequences (Fig. 7). However some of this defects such as the altered mechanical coupling between cytoskeleton and NE, and the YAP cytoplasmic retention occur soon after ATR catalytic inactivation, suggesting that these two phenotypes represent a direct consequence of ATR deregulation. Considering that during development, organogenesis requires that stem/progenitor cells migrate towards destination tissues, our observations may contribute to explain some of the developmental defects of Seckel patients bearing genetic defects in ATR. Our results might also explain the increased cell death in non-proliferating neuroprogenitors and neurons of ATR-knockout mice[51,52], which cannot be directly ascribed to the role of ATR in replication stress. Moreover, the delocalization of YAP might also contribute to a variety of pathological outcomes such as loss of stem cells and cardiomyopathies[27,53]. Intriguingly, ATR-conditional knockout mice exhibit a progeroid phenotype that has been related to stem cell loss[54].

## Methods

**Plasmids**. ATR shRNA1 and control (pLKO1) plasmids were from Dr. O.F. Capetillo (CNIO, Spain); ATR shRNA2 was purchased from Sigma (TRCN0000219647); the GFP-ATR plasmid was from Dr. R.Tibbetts (Wisconsin, USA)[55]. GFP-AU1-ATR plasmid was digested with BamHI to excise out GFP cDNA. The BamHI-digested GFP cDNA insert was cloned into FLAG-ATR-KD plasmid[56] also linearized with BamHI, followed by transformation in *Escherichia coli*. Positive clones after transformations were screened by PCR and BamHI restriction digestion, finally sequenced (list of primers used are provided in Supplementary Table 1). pTRIP-CMV-GFP-FLAG-cGAS (Plasmid #86675)[28], Nesprin tension sensor (pcDNA nesprin TS; plasmid #68127), and nesprin headless control (pcDNA nesprin HL; plasmid #68128)[25] were acquired from Addgene plasmid repository.

| Antibody | Source | Catalog number | Dilution |
|---|---|---|---|
| 1. ATR | Cell Signal | 2790 | 1 : 1000 (WB); 1 : 100 (IF) |
| 2. TopBP1 | Abcam | ab2402 | 1 : 1000 (WB) |
| 3. Nup133 | SantaCruz | sc-27392 | 1 : 500 (WB) |
| 4. Tubulin | Sigma | T5168 | 1 : 5000 (WB) |
| 5. Lamin B1 | Abcam | ab16048 | 1 : 10,000 (WB); 1 : 1000 (IF) |
| 6. Lamin A/C | SantaCruz | sc-7292 | 1 : 500 (WB); 1 : 200 (IF) |
| 7. Nesprin 2 | Thermo Scientific | MA5-18075 | 1 : 500 (WB); 1 : 500 (IF) |
| 8. Histone H3(tri-met K9) | Abcam | ab8898 | 1 : 1000 (WB) |
| 9. Total Histone H3 | Abcam | ab1791 | 1 : 5000 (WB) |
| 10. Total YAP(63.7) | SantaCruz | sc-101199 | 1 : 500 (WB); 1 : 200 (IF) |
| 11. Phospho YAP (Ser127) | Cell Signal | 4911S | 1 : 1000 (WB) |

Secondary antibodies were obtained from IFOM imaging facility:
Polyclonal Donkey anti-mouse AlexaFluor-488 AB_2340846 (Jackson ImmunoResearch).

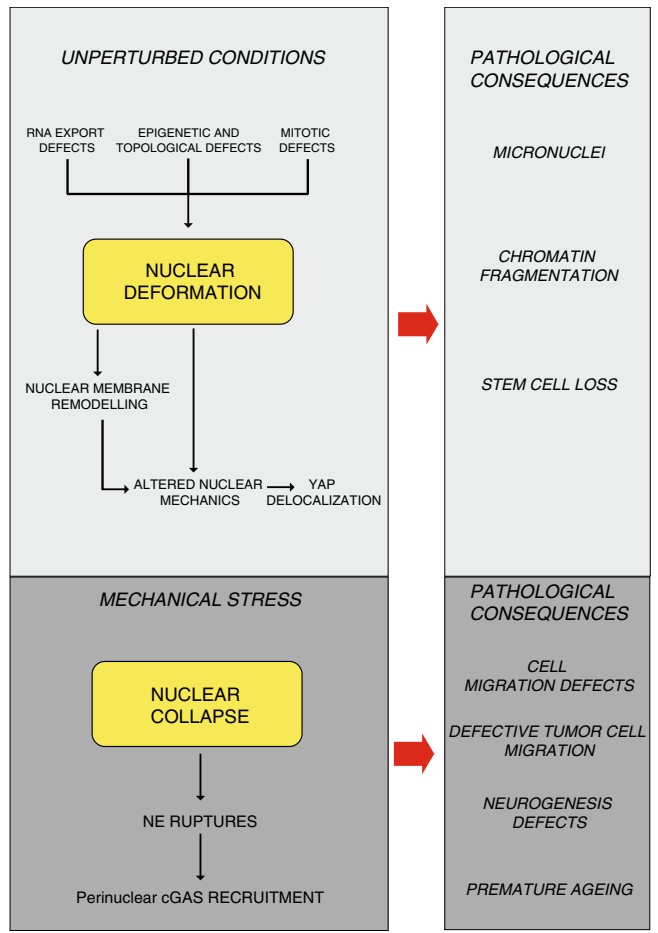

**Fig. 7 Graphical summary of ATR defects affecting nuclear morphology and mechanics and relative pathological consequences.** In the absence of external stimuli ATR coordinates chromatin processes (such as RNA export, epigenetic and topological transitions, and chromatin condensation in prophase) with NE dynamics, influencing nuclear morphology. ATR defects lead to nuclear deformation, NE remodeling, altered nuclear mechanics, and YAP delocalization. In response to mechanical stress, ATR-defective nuclei collapse leading to NE ruptures and cGAS recruitment at the nuclear periphery. Pathological consequences are also described. See text for details.

Polyclonal Donkey anti-mouse AlexaFluor-594 AB_2340854 (Jackson ImmunoResearch).

Polyclonal Donkey anti-mouse AlexaFluor-Cy3 AB_2340813 (Jackson ImmunoResearch).

Polyclonal Donkey anti-rabbit AlexaFluor-488 AB_2313584 (Jackson ImmunoResearch).

Polyclonal Donkey anti-rabbit AlexaFluor-594 AB_2340621 (Jackson ImmunoResearch).

Polyclonal Donkey anti-rabbit AlexaFluor-Cy3 AB_2307443 (Jackson ImmunoResearch).

| Other reagents | Source | Catalog number |
|---|---|---|
| 1. ATR inhibitors | | |
| ETP46464 | Calbiochem | 500508 |
| AZ-20 | Tocris | 5198 |
| VE-821 | Selleckchem | S8007 |
| 2. Benzonase | Sigma | E1014 |
| 3. Anti-GFP mAb-Magnetic beads | MBL, Japan | D153-11 |
| 4. Mem-PER Plus Kit | Thermo Fisher | 89842 |
| 5. Duolink® PLA kit | Sigma | DUO92102 |
| 6. BCA Protein Assay | Thermo Fisher | 23227 |
| 7. RTV615 (PDMS) | Momentive Perf. Materials | RTV615 |
| 8. Leptomycin-B | Sigma | L2913-.5UG |
| 9. DNAse I | Promega | M6101 |

**Cell lines**. U2OS cells stably expressing GFP-ATR and HeLa cells stably expressing mCherry-H2B were reported previously[6]. U2OS cells expressing the FUCCI reporter was a kind gift from Libor Macůrek[33]. Human primary Seckel fibroblasts (GM18366) and IMR90 were from Coriell Cell Repository. HCT116 and ATR$^{flox/-}$ cells were from The American Type Culture Collection.

**Cell culture, transfection, and inhibitor treatments**. HeLa and U2OS cells were maintained in Dulbecco's modified Eagle's medium (DMEM) with GlutaMAX (Life Technologies) supplemented with 10% (vol/vol) fetal bovine serum (FBS, Biowest) and penicillin–streptomycin (Microtech). Human primary fibroblasts derived from Seckel patient were maintained in DMEM supplemented with 15% FBS (not activated, Sigma-Aldrich) and IMR90 were grown in 10% FBS (not activated). HCT116 and ATR$^{flox/-}$ cells were grown in McCOY's 5A media. All cells were grown in a humidified incubator atmosphere at 37° and 5% $CO_2$.

We used Lipofectamine 2000 (Invitrogen) for transfecting plasmids into cells, using the protocol recommended by the manufacturer.

HEK293T cells were transfected with shRNA plasmids and viral packaging plasmids to generate lentiviral particles. Desired cell lines were then infected for 16 h followed by 2 µg/ml puromycin selection for 24 h. Infected cells were cultured in 1 µg/ml puromycin containing media and were utilized for experiments up to 10 days after infection.

Cells were treated with ATR inhibitors (2 µM ETP46464, 10 µM VE-821, or 1 µM AZ-20) 1 h before (unless mentioned otherwise) starting the experiment and were maintained in the media throughout the course of the experiment.

For cell cycle analysis, cells were fixed with ice-cold ethanol, DNA was labeled with propidium iodide, and quantified using FACS calibur (BD bioscience) system.

**Membrane fractionation using Mem-Per Plus kit**. Membrane fractionations were performed following protocol provided by the vendor. Briefly, cells were trypsinized, washed with cell wash solution, resuspended in permeabilization buffer (with or without Benzonase), and incubated for 30 min at 4 °C with constant mixing. Permeabilized cells were then centrifuged for 15 min at 16,000 × g, soluble fraction was collected, and pellet resuspended and incubated in solubilization buffer for 30 min at 4 °C. Samples were then centrifuged for 15 min at 16,000 × g and supernatant was collected as a membrane fraction.

**Cell lysis and immunoblotting**. Total cell lysates were prepared in lysis buffer (50 mM Tris-HCl pH 8.0, 1 mM $MgCl_2$, 200 mM NaCl, 10% Glycerol, 1% NP-40) Protease (Roche) and Phosphatase inhibitors (Sigma) were added at the time of experiment, and Benzonase (50 U/ml) was added if degradation of nucleic acid was needed. Cell lysates boiled with Laemmli buffer were (20–50 µg) resolved using NuPAGE® (Invitrogen) or Mini-PROTEAN® (Biorad) precast gels, transferred to nitrocellulose membrane, and probed as with primary (2 h at roomtemperature (RT) or overnight at 4 °C) and secondary antibodies (1 h at RT), and acquired using ChemiDoc imaging system (Image Lab v5.0). Image intensity measurements were performed using ImageJ.

**IF assays and quantifications**. Briefly, cells were fixed with 4% formaldehyde (15 min), permeabilized with 0.2% Triton X-100 in phosphate-buffered saline (PBS) (15 min), blocked with 1% bovine serum albumin in PBS for 1 h (blocking buffer), incubated with primary antibodies (diluted in blocking buffer) for 1 h in RT, followed by three PBS washes and then incubated in secondary antibodies (1 : 400 in blocking solution) for 1 h in the dark at RT followed by three PBS washes. Samples were mounted with VectaShield mounting medium containing 4′,6-diamidino-2-phenylindole (DAPI). Image acquisition was performed using Leica TCS SP2 confocal scanning microscope, equipped with a ×63/1.4 numerical aperture (NA) objective. Single optical sections of the images or maximum projections (step size 0.5 µm) were processed using ImageJ and smoothed to reduce the background noise.

*Quantification of nuclear morphology, YAP localization*: images from random fields (upto 50) were acquired from coverslips stained with DAPI and Lamin or YAP on a UltraVIEW VoX spinning-disc confocal unit with Velocity software (PerkinElmer). Nuclei from each field were manually binned into normal, mild (blebs, invaginations, wrinkles, micronuclei, and multi-nuclei), as well as severely deformed (with multiple defects), as well as with or without micronuclei alone (in case if field has <20 cells it is combined with next field), which then are averaged to perform statistical analysis. Circularity index was calculated on central section of Lamin staining using ImageJ particle analysis tool and ABsnake plugin. YAP localization analysis was performed following the method described in Elosegui-Artola et al.[26]. Ratio was calculated between gross intensity measurements from a circular region of 30 pixel diameter in the nucleus and in the cytoplasm from individual cell.

**Electron microscopy**. The staff of EM facility at IFOM performed all the EM analysis. EM examination, Immuno-EM gold labeling based on pre-embedding, EM tomography, and correlative light-electron microscopy (CLEM) were

performed exactly as it has been reported previously[6,57,58]. A brief description of each process is described below.

*Embedding*: cells grown on MatTek dishes (MatTek Corporation, USA) were fixed with of 4% paraformaldehyde and 2.5% glutaraldehyde (EMS, USA) mixture in 0.2 M sodium cacodylate buffer (pH 7.2) for 2 h at RT, followed by six washes in 0.2 M sodium cacodylate buffer at RT. Then cells were incubated in 1 : 1 mixture of 2% osmium tetraoxide and 3% potassium ferrocyanide for 1 h at RT followed by six times rinsing in 0.2 M sodium cacodylate buffer (pH 7.2). Then the samples were sequentially treated with 0.3% Thiocarbohydrazide (in 0.2 M sodium cacodylate buffer) for 10 min and 1% OsO4 (in 0.2 M cacodylate buffer, pH 6.9) for 30 min. Samples were then rinsed with 0.1 M sodium cacodylate (pH 6.9) buffer until all traces of the yellow osmium fixative have been removed. Then samples were washed in de-ionized water, treated with 1% uranyl acetate (in distilled water) for 1 h and washed in water again[57,59]. The samples were subsequently embedded in Epoxy resin at RT and polymerized for at least 72 h in a 60 °C oven. Embedded samples were then sectioned with diamond knife (Diatome, Switzerland) using the ultramicrotome (LeicaEM UC7, Leica Microsystem, Vienna). Sections were analyzed with a Tecnai20 EM (FEI, Thermo Fisher Scientific, Eindhoven, The Netherlands) operating at 200 kV[58].

*Nano-gold labeling*: cells grown on MatTeks were fixed with a mixture of 4% paraformaldehyde and 0.05% glutaraldehyde (0.15 M Hepes buffer, pH 7.2) for 5 min at RT and then replaced with 4% paraformaldehyde (in 0.15 M Hepes buffer, pH 7.2) for 30 min. Cells were washed six times in PBS and incubated with blocking solution for 30 min at RT. Then cells were incubated with primary antibody diluted in blocking solution overnight at 4 °C. On the following day, the cells were washed six times with PBS and incubated with goat anti-rabbit Fab' fragments coupled to 1.4 nm gold particles (diluted in blocking solution 1 : 100) for 2 h and washed six times with PBS. Meanwhile, the activated GoldEnhanceTM-EM was prepared according to the manufacturer's instructions and 100 μl of it was added into each sample well. The reaction was monitored by a conventional light microscope and was stopped after 5–10 min when the cells had turned "dark enough" by washing several times with PBS. Osmification followed: the cells were incubated for 1 h at RT with a 1 : 1 mixture of 2% osmium tetraoxide (in water) and 3% potassium ferrocyanide (in 0.2 M sodium cacodylate pH 7.4) and then rinsed six times with PBS and then with distilled water. The samples were then dehydrated: 3 × 10 min in 50% ethanol; 3 × 10 min in 70% ethanol; 3 × 10 min in 90% ethanol; 3 × 10 min in 100% ethanol. The samples were subsequently incubated for 2 h in 1 : 1 mixture of 100% ethanol and Epoxy resin (Epon.EMS) at RT; the mixture was then removed with a pipette and finally samples were embedded for 2 h in Epoxy resin at RT. The resin was polymerized for at least 10 h at 60 °C in an oven.

*Tomography*: two-step CLEM based on the analysis of tomographic reconstructions acquired under low magnification with consecutive reacquisition of EM tomo box under high magnification (×60,000) and its re-examination was used exactly as described previously[60]. Briefly, an ultratome (LeicaEM UC7; Leica Microsystems, Vienna) was used to cut 60 nm serial thin sections and 200 nm serial semi-thick sections. Sections were collected onto 1% Formvar films adhered to slot grids. Both sides of the grids were labeled with fiduciary 10 nm gold (PAG10, CMC, Utrecht, The Netherlands). Tilt series were collected from the samples from ±65° with 1° increments at 200 kV in Tecnai20 EMs (FEI, Thermo Fisher Scientific, Eindhoven, Tthe Netherlands). Tilt series were recorded at a magnification of ×20,000 or ×60,000 using software supplied with the instrument. The nominal resolution in our tomograms was 4 nm, based upon section thickness, the number of tilts, tilt increments, and tilt angle range. The IMOD package and its newest viewer, 3DMOD 4.0.11, were used to construct individual tomograms and for the assignment of the outer leaflet of organelle membrane contours, and best-fit sphere models of the outer leaflet were used for vesicle measurements. Videos were made in 3DMOD and assembled in QuickTime Pro 7.5 (Apple) and the video size was reduced by saving videos at 480p in QuickTime. CLEM was performed exactly as described previously[59].

*FIBSEM*: FIBSEM analysis was performed using a FEI Helios NanoLab 660 FEGSEM or G3 equipped with SEM Multi-Detector and ICD detector at accelerating voltage 2.0 kV. Access to both of which was kindly provided by FEI, Co. (FEI, Thermo Fisher Scientific, Eindhoven, The Netherlands). For all high-resolution EFSEM images, a primary beam energy of 2.0 kV was used with a working distance of 1 mm, 3 ms dwell time, and tube bias of 140 V. An Auriga 60 FIB-SEM (Zeiss) microscope with Atlas3D software (FIBICS) was additionally used to collect the 3D data of two cells. Acquisitions were performed according to instructions of the manufacturers.

*Quantification and statistics*: all acquired images were aligned using the TrakEM2 plugin of FIJI. Images were segmented by thresholding with Amira ((FEI, Thermo Fisher Scientific, Eindhoven, The Netherlands). The number of ATR-tagged gold particles in different compartments of the cell was counted and percentages were calculated. The labeling density of ATR on different cellular structures was assessed and calculated as described in ref. [61]. For this we used the following criteria: gold particles were considered to label the NE, ER, or mitochondria when these particles were observed over lumens or membranes of these compartments; gold particles were considered as a label of the PM when these particles were observed over the PM. Normality of variant distribution was assessed with Shapiro–Wilk tests. Cumulative probability distributions were compared

using the Kolmogorov–Smirnov test. Estimation of the minimal set of samples was performed according to ref. [62]. Correlation between two variables was calculated using Pearson's product moment correlation.

*Analysis of cells in channels*: we embedded Poly-di-methyl-siloxane (PDMS) molds on MatTek dishes, loaded cell, and incubated for 24 h to facilitate cell migration into the channels. Cells migrating within the channels were examined under the UltraVIEW VoX spinning-disc confocal unit (PerkinElmer) and acquired images of cells suitable for the future CLEM analysis. The remaining cells from the loading wells were eliminated and 0.05% glutaraldehyde + 4% formaldehyde solution (in 0.1 M cacodylate buffer, pH 7.2) was added to the dish for 5 min. Cells were then fixed with 2.5% glutaraldehyde + 4% formaldehyde (in 0.2 M cacodylate buffer, pH 7.2) for 10 days, to make cell bodies resistant to the process of the mechanical detachment of PDMS from the MatTek. Then PDMS mold was detached from the MatTek dish and the cells attached to the dishes were processed for EM analysis as described above. After mold detachment, cells were additionally stained with 1% methylene blue in PBS for 3 min at RT and again examined under a light microscope to confirm the presence of selected cells on the MatTek glass.

**DNAse I sensitivity assay**. Cells were trypsinized, washed in ice-cold PBS, and resuspended in 2 ml of ice-cold cell lysis buffer (300 mM sucrose, 10 mM Tris pH 7.4, 15 mM NaCl, 5 mM MgCl, 0.5% NP-40, 0.5 mM dithiothreitol, protease inhibitor (Complete, Roche)). After 30 min, the lysed cells were centrifuged at 500 × g for 5 min at 4 °C and supernatant was discarded. The nuclei were gently resuspended in appropriate amount of reaction buffer (30 μl per DNAse I condition). Separate 30 μl aliquots were then taken and gently mixed with 70 μl of DNase I mix (of varying units) on ice. It was incubated for 15 min at 25 °C and then 700 μl of nuclei lysis buffer (100 mM Tris-HCL pH 8, 5 mM EDTA pH 8, 200 mM NaCl, 0.2% SDS) was added to each sample with 50 mg proteinase K. Samples were incubated at 55 °C for 1 h; RNaseA (10 mg) was added and again incubated at 37 °C for 30 min. DNA was then extracted using standard phenol–chloroform technique and was resuspended in 200 μl of 0.1 TE, quantified, and were run on 1% agarose gels.

**AFM measurements**. The AFM measurements were performed using Nanowizard 3 (JPK Instruments, Germany) and a modified silicon nitride cantilever (NovaScan, USA) with a spring constant of 0.03 N/m and a 5 μm diameter polystyrene bead adhered at the tip. Central region of the cell was indented with a loading rate of 1.5 μm/s. The ramp size was 3 μm was used. All the measurements were performed as previously described in ref. [17]. Nuclei were isolated by treating cells with 1 ml of a 0.01% Igepal CA-630 (a non-ionic detergent, Sigma) and 1% citric acid solution in water for 5 min. Expelled nuclei from the adherent cells were collected, washed with 5 ml PBS, centrifuged at 800 × g for 5 min, resuspended in PBS, and dropped onto coverslip for AFM experiments.

**FRET image acquisition and analysis**. Cells grown on coverslips were injected with the Nesprin 2G-TS construct (50 ng/cell) and the following day, imaged using a DeltaVision Elite imaging system using an Olympus ×60/1.42 Plan Apo N oil-immersion objective. Three images were collected in sequence at each point: Cyan Fluorescent Protein (CFP) (for mTFP1) (ex: 438/24 nm, em: 470/24 nm), FRET (ex: 438/24 nm, em: 559/38 nm), and Yellow Fluorescent Protein (for mVenus) (ex: 513/17 nm, em: 528/38 nm). A single-plane image was background corrected, realigned, converted into 32 bits, and analyzed using an in-house macro in ImageJ. The nuclear membrane of each cell was manually selected as a region of interest and average FRET/CFP ratios calculated for the nuclear membrane region. Approximately 30 cells/conditions were analyzed for $n = 2$ experiments. ATR inhibitor was added (ETP46464 2 μM) 3 h before image acquisition. ATR inhibitor VE-821 could not be used for this measurements, as they exhibited auto fluorescence. HeLa cells stably expressing Nesprin 2G sensor or the Headless control sensor were generated by Lipofectamine 2000 transfection, Neomycin (G418) selection, and single-cell fluorescence-activated cells sorting of the mVenus/mTFP1-positive population. These cells were loaded onto the channels in the presence of ATR inhibitor or DMSO. Single stack image acquired for each field of view every 2 h for 10 h duration. Image acquisition parameters and analysis were similar to the above-mentioned experiment.

**FLIM-FRET analysis**. For the acquisition, we used the Leica TCS SP8 confocal microscope with White light laser as excitation source tuned at 488 nm and HC PL APO CS2 ×63/1.40 oil-immersion objective, everything managed by Leica Application Suite X software, ver. 3.5.2.18963. For the lifetime measurements, the above system was implemented with PicoQuant Pico Harp 300 TCSPC module and picosecond event timer, managed by PicoQuant software (SymPho Time 64, ver. 2.4). Data were imported and analyzed using in-house ImageJ macro.

**Micro-fabricated cell compression chamber**. A custom-made cell compression device has been invented based on movement of thin membrane attached with a piston, which is precisely controlled by air pressure regulator. The cell compression device was designed using Solidworks and device components were 3D-printed using Dental SG resin (Formlabs) for its biocompatibility. All the components were

printed and then washed with IPA for 20 min, followed by post processing in ultraviolet chamber as suggested by Formlabs. A 20 mm diameter coverslip was stick on the top center of the cell compression device. Silicon membrane was sticked with a piston and then clamped to the bottom of the cell compression device by clamping tools. The assembled cell compression device was then connected to the air pressure regulator. Cells were plated on glass-bottom petridish and maintained in cell incubator. Before the experiment, a cell compression device was capped and locked on the cell culture dish. Images were acquired using ×40 oil lens (NA = 1.3) in PerkinElmer spinning disk microscope.

**Analysis of cell migration in micro-fabricated channels**. We followed the protocol established previously[28] for PDMS channels preparation. Briefly, Polymer and crosslinking agent (RTV615 kit) (mixed in 1 : 10 ratio) was used to prepare PDMS channels. These are then plasma treated and embedded onto a glass-bottom dish or a two-chamber LAB-TEK II dish (Thermo Fisher). Then channels were fibronectin-coated and cells were loaded the day before the beginning of time lapse. We chose 15 μm-long, 4 μm-wide constriction for experiments involving HeLa and U2OS cell lines. Time-lapse images were acquired (every 10 or 15 min, with z-stacks) on a UltraVIEW VoX spinning-disc confocal unit with Velocity software (PerkinElmer), equipped with an Eclipse Ti inverted microscope (Nikon) and a C9100-50 electron-multiplying CCD (charge-coupled device) camera (Hamamatsu) or Confocal Spinning Disk microscope (Olympus) equipped with IX83 inverted microscope provided with an IXON 897 Ultra camera (Andor) with OLYMPUS cellSens Dimension software, or on a DeltaVision Elite imaging system using ×40 oil-immersion (for 53BP1 foci counting) or ×20 dry objective for a duration of 18–24 h. The images were processed using ImageJ and smoothed to reduce the background noise. All the quantifications were performed manually. Number of cells reaching the constriction within the experimental period was considered as a total cell number. Number of cell death and cell passing were counted per field to calculate the percentages. Fields with no cell migration or death were discarded from analysis. 53BP1 foci were counted manually using ImageJ. Difference was calculated by subtracting number of Foci before engaging the constriction from the number of foci present in the constriction.

**IP, MS, and data analysis**. U2OS cells expressing GFP-ATR or GFP alone are cultured in SILAC medium containing light or heavy-labeled L-lysine and L-arginine for 5 days ensuring adequate incorporation of isotopes. These cells were collected with Lysis buffer (50 mM Tris-HCl pH 8.0, 1 mM MgCl₂, 200 mM NaCl, 10% Glycerol, 1% NP-40 + Protease, and Phosphatase inhibitors) containing Benzonase (50 U/ml) and incubated for 1 h on ice. Lysates were pre-cleaned by incubating with Protein-A beads for 1 h at 4 °C. GFP and GFP-ATR (4 mg of protein lysate per sample) were immunoprecipitated by incubating the lysates with 200 μl of anti-GFP-conjugated magnetic beads overnight on a rotor at 4°. Beads were washed (with Lysis buffer) and eventually pooled before elution with sample buffer. Proteins were then resolved onto a 4–12% NuPAGE® precast gel (Invitrogen) and stained by Coomassie colloidal blue. The gel lane was cut into eight or ten slices each of which has been reduced, alkylated, and digested with trypsin as reported elsewhere[63]. Peptide mixtures were desalted and concentrated on a homemade C18 desalting tip, then peptides were injected in a nanoHPLC (EasyLC Proxeon, Denmark). Peptides separation occurred onto a 25 cm-long column, reverse-phase spraying fused silica capillary column (75 μm i.d.) packed with 3 μm ReproSil AQ C18 (Dr. Maisch GmbH, Germany). A gradient of eluents A (high-performance liquid chromatography-grade water with 0.1% v/v formic acid) and B (Acetonitrile) with 20% v/v water with 0.1% v/v formic acid) were used to achieve separation, from 7 to 60% of B in 30 min, at a constant flow rate of 250 nl/min. The LC system was connected to a QExactiveHF mass spectrometer (Thermo Scientific, Bremen, Germany) equipped with a nanoelectrospray ion source (Proxeon Biosystems, Odense, Denmark). Full scan mass spectra were acquired in the LTQ Orbitrap mass spectrometer with the resolution set to 60,000 (@200 m/z) accumulating ions to a target value of 6,000,000. The acquisition mass range for each sample was from m/z 300–1650 Da and the analyses were made in duplicates. The 15 most intense doubly and triply charged ions were automatically selected and fragmented in the ion trap after accumulation to a "target value" of 15,000. Target ions already selected for the MS/MS were dynamically excluded for 20 s. Identification and quantification of peptides and proteins were performed with MaxQuant 1.5.2.8 against the human Uniprot complete proteome set, having identified a protein with at least two peptides (one unique), six amino acids of minimal length, false discovery rate <1%, and quantified with at least two ratio counts. Significant outliers scores were calculated using Perseus 1.5.2.6[64] and those with a p-value < 0.05 have been selected for further analysis. For label-free analysis, the procedure was the same but immunoprecipitated samples were kept separate and loaded separately, then digested and analyzed by MS. Proteins were identified using Mascot (v. 2.3.02) and quantification was done using Scaffold (v. 4.3.4). Exclusive unique peptide count was selected to evaluate changes in proteins abundance.

For analysis, we pooled data from all experiments, selected candidates that were found as ATR-GFP interactors in at least two experiments, and which were significantly enriched (fourfold) over GFP control in at least one SILAC experiment. We performed Gene Ontology (GO) analysis using DAVID, to generate enriched terms for cellular compartments (GOTERM_CC) and biological

processes (GOTERM_BP) (p-value with Benjamini correction < 0.05). Revigo tool was utilized to simplify GO terms (resulting list size: 0.7, database: Homo sapiens, semantic similarity measure: SimRel) and R Studio for plotting of Revigo output (size = log size, color = log10 p-value). Candidates where then manually curated to generate non-overlapping sub-categories of interest (for this study). A network was generated for each sub-category using STRING interaction analysis and the output was plotted using cytoscape.

The MS proteomics data have been deposited to the ProteomeXchange Consortium via the PRIDE[65] partner repository with the dataset identifier PXD020622 (Project Name: ATR interactome in H. sapiens bone osteosarcoma U2OS cells). Files are named SILAC1, SILAC2, and SILAC3.

**Lipidomic analysis**. (I) Lipid extraction: Nuclei were isolated according to the protocol[66]. Nucleus and total cell samples were resuspended in 150 mM ammonium bicarbonate and passed through a 26 G syringe needle to fragment nucleic acids. Samples were centrifuged at 10,000 × g for 10 min at 4 °C to eliminate cell debris. Lipids were extracted starting from an sample size equivalent of 50 μg of proteins, using a two-step extraction protocol (Folch method) with methanol and chloroform in different proportions[67]. Organic phase fractions were then dried out and resuspended in 50 μL of 95% phase A (CH₃CN : H₂O 40 : 60; 5 mM NH₄COOCH₃; 0.1% FA) plus 5% phase B (IPA : H₂O 90 : 10; 5 mM NH₄COOCH₃; 0.1% FA) for subsequent analysis. Before extraction, samples were spiked in with 16 internal standards: PC (12 : 0/13 : 0) 40 pmol, PE (12 : 0/13 : 0) 52 pmol, phosphatidylglycerol (PG) (12 : 0/13 : 0) 7.5 pmol, phosphatidylserine (PS) (12 : 0/13 : 0) 43 pmol, phosphatidylinositol (PI) (12 : 0/13 : 0) 54 pmol, Cer (d18 : 1/25 : 0) 100 pmol, cholesterol ester (CE) (19 : 0) 100 pmol, GlcCer (d18 : 1/12 : 0) 50 pmol, LacCer (d18 : 1/12 : 0) 50 pmol, sphinganine (d17 : 0) 50 pmol, sphingosine-1-P (d17 : 1) 100 pmol, sphingosine (d17 : 1) 50 pmol, Galactosyl(β) Sphingosine-d5 20 pmol, d5-TG ISTD Mix I 20 pmol, d5-DG ISTD Mix I 20 pmol, and cholesterol (d7) 800 pmol. (II) Protein quantification: proteins were extracted form 20 μL of ammonium bicarbonate resuspended fractions by adding 5 μL of lysis buffer (10% NP-40, 2% SDS in PBS) and quantified by BCA protein assay kit (Thermo Fisher Scientific). (III) Lipid profiling data acquisition: lipid extracts were diluted 1 : 5 and 1 μL injected on a LC system nLC Ekspert nanoLC400 (Eksigent, 5033460C; Singapore) coupled with a Triple TOF 6600 (AB Sciex, Singapore). Chromatography was performed using an in-house packed nanocolumn Kinetex EVO C18, 1.7 μm, 100 A, 0.75 × 100 mm. The mobile phases A (CH₃CN : H₂O 40 : 60; 5 mM NH₄COOCH₃; 0.1% FA) and B (IPA : H2O 90 : 10; 5 mM NH₄COOCH₃; 0.1% FA) were used in positive mode. The gradient elution was initially started from 5% B, linearly increased to 100% B in 5 min, maintained for 45 min, then returned to the initial ratio in 2 min, and maintained for 8 min. Acquisition in MS was performed in positive with the following parameters: mass over charge (m/z) range 100–1700, T source 80 °C, Ion Spray Voltage 2000, declustering potential 80, fixed collision energy 40 V (+). For Information Dependent Acquisition analysis (Top 8), range of m/z was set as 200–1800 in positive ion mode; target ions were excluded for 20 s after two occurrences (Analyst TF 1.7.1). (IV) Data processing: Lipidview workstation (version 1.3 beta, AB SCIEX, USA) was for lipids identification and quantification. Lipid identification was based on exact mass, retention time, and MS/MS pattern. Lipid species based on precursor fragment ion pairs were determined using a comprehensive target list in LipidView (Sciex). Lipid species identification was performed using the mass tolerance of 0.05 in MS and 0.02 in MS/MS, s/n of 3, and % peak intensity >0 for positive ion mode. Lipid classes included for statistics and downstream analysis were cholesterol ester (CE), sphingomyelin, diacylglycerol, triacylglycerol, ceramide (Cer), PC, PE, PG, PI, PS and lysophosphatidylcholine, lysophosphatidylethanolamine, lysophosphatidylglycerol, lysophosphatidylinositol, lysophosphatidylserine, hexosylceramide, dihexosylceramide, trihexosylceramide, sulphatides (SGalCer), and Cer-phosphate, in positive mode. All statistical analysis was performed using Metaboanalyst 4.0 web tool[68] (https://www.metaboanalyst.ca/MetaboAnalyst/faces/home.xhtml). Three experimental and two technical replicates were measured for each condition and 854 lipid metabolites were detected in total. Missing metabolite intensities were imputed with half of the lowest detectable value. Intensities were then normalized by the median of the 322 technically most reliable metabolites (detectable in ~95% of all samples) and averaged over two technical replicates. The total intensities of PC and PE were calculated as sum of all individual phospholipids with choline and ethanolamine head groups.

**Proximity ligation assay**. Experiment was performed using rabbit polyclonal ATR antibody (1 : 100) and mouse monoclonal Nesprin-2 antibody (1 : 250), following the protocol from the manufacturer (Sigma, Duolink PLA Technology).

**Animals**. Wild-type B6/CBAF1 mice used for in utero electroporation were purchased from The Jackson Laboratory. Mice lines (Atr-CER, 129/Sv, and C57BL/6 mixed background) with inducible deletion of Atr were generated as reported in ref. [51]. All animals were maintained in the Specific Pathogen-Free facility and experiments were conducted according to German animal welfare legislation. All animals were housed at a temperature 22 ± 2 °C, humidity 30–70%, and 12 h/12 h

dark/light cycle. Animal experiment protocol was approved by Thüringen Land-esamt für Verbraucherschutz, Germany.

Primers used for genotyping are as follows:
ATR10 (5′-CTATTTTTTGTTGCTGGTTTTG-3′)
ATR15 (5′-CTTCTAATCTTC-CTCCAGAATTGTAAAAGG-3′)
Cre1 (5′-CGGTCGATGCAACGAGTGATG-3′)
Cre2 (5′-CCAGA-GACGGAAATCCATCGC-3′).

**Transwell membrane migration assay.** Neuroprogenitors were isolated from E13.5 embryonic brain of Atr-inducible deletion (ATR-CER) mouse line brain and cultured in neurosphere medium (DMEM/F12 supplemented with B-27, penicillin/streptomycin, 10 ng/ml epidermal growth factor (EGF), and 20 ng/ml basic fibro-blast growth factro (bFGF)) for 1~2 days. Then 4-hydroxytamoxifen (4-OHT) was added to induce Atr deletion. Four days after tamoxifen or 4-OHT treatment, neurosphere were trypsinized and resuspended in EGF- and bFGF-free medium. Cells were plated into polycarbonate membrane insert (ThinCert™, Greiner-Bio-One GmbH, Frickenhausen Germany) coated with poly-L-lysine. The cells were allowed to migrate for 20 h at 37 °C in 5% $CO_2$ before fixation with paraf-ormaldehyde and staining with DAPI. All cells in the upper side of membrane were carefully removed with a cotton ball before mounting the membrane to glass slice with coverslip on the top. Cells on the underside of filter membrane were counted under a fluorescence microscope and migration activity was calculated by average number of cells per object (×20) field.

**In vivo neuronal migration.** The construction of shRNA expression vectors was reported in ref. [69]. All oligonucleotides contained the hairpin loop sequence 5′-TTCAAGAGA-3′. The targeting sequences are as follows:
shLuc: 5′-GGCTTGCCAGCAACTTACA-3′;
shATR-4: 5′-GGACCTAAACATGTCAGTTCT-3′;
shATR-6: 5′-GCATGCCATCAGTACCCAAGA-3′.
The efficiency of these shRNA was screened with mouse embryonic fibroblasts cells and Neuro 2A cells. In utero electroporation was performed as described in ref. [70]. One microgram of plasmid DNA in PBS was injected into the lateral ventricle of E14.5 embryos followed by electroporation. The embryos were isolated 4 days after electroporation (at E18.5) and processed for cryo-section. Images were acquired with Zeiss ApoTome (Carl Zeiss, Germany) after immunostaining with DAPI. GFP-tagged shRNA vectors were electroporated into wild-type embryonic brain ventricles at E14.5. The embryonic brain is analyzed by imaging at E18.5. Brain cortex was equally divided into ten segmentations and percentage of GFP-positive (GFP+) cells from each segmentation were quantified based on 1~2 sections from indicated number of animals for each plasmid.

**Short-term lung colonization assay.** Control HeLa cells ($5 \times 10^5$) and shATR HeLa cells ($5 \times 10^5$) were labeled with E-Fluor 670 (Molecular Probes), mixed in 200 µl PBS, and injected intravenously. Mice were then sacrificed after 2 and 48 h. The lungs were isolated and fixed in 4% phosphate-buffered formalin. Micrometastases were visua-lized using a confocal microscope and counted. All animal experiments were approved by the OPBA (Organisms for the well-being of the animal) of IFOM and Cogentech. All experiments complied with national guidelines and legislation for animal experimentation. All animal experiments were performed in accordance with national and international laws and policies. Mice were bred and housed under pathogen-free conditions in our animal facilities at Cogentech Consortium at the FIRC Institute of Molecular Oncology Foundation under the authorization from the Italian Ministry of Health (Autorizzazione Number 604-2016).

**Statistics and reproducibility.** Statistical calculations and graphs were generated with GraphPad Prism5 and Prism7 or using Microsoft Excel (2011) software. All bar graphs are represented as mean ± s.e.m. Box plot, whiskers, and outliers are plotted in Graphpad Prism7 using Tukey's method. Each dot on the box plots represents a measurement from a single cell. P-values were calculated by Student's t-test, one-way or two-way analysis of variance with Sidek's, Tukey's, Dunnett's, or Bonferroni multiple comparisons as indicated in the figure legends of the respective figure. All detailed report of statistical analysis for each graph is included in the source data file. All the experiments presented in this manuscript are successfully reproduced at least in two independent experiments. Exact numbers of replicates are included in the figure legend.

**Reporting summary.** Further information on research design is available in the Nature Research Reporting Summary linked to this article.

## Data availability
The mass spectrometry proteomics data have been deposited to the ProteomeXchange Consortium via the PRIDE[65] partner repository with the dataset identifier PXD020622. All other data supporting the findings of this study are available in main text, the Supplementary Material or in Source Data file. Any additional information can be made available personally upon reasonable request to the corresponding author. Source Data are provided with this paper.

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

## Acknowledgements

We thank O. Capetillo (CNIO, Spain), R. Tibbetts (Wisconsin) for reagents, Y. Schwab and J.M. Serra Lleti (EMBL, Germany), Letian Li and Anthony Burgess (Thermo Fisher Scientific, The Netherlands) for assistance in EM analysis, Michele Giannattasio and the members of MF and GS laboratories for assistance and suggestions. We thank IFOM cell culture and imaging facility and Cogentech sequencing services for technical assistance. Work in MF's laboratory is supported by grants from Associazione Italiana per la Ricerca sul Cancro, AIRC, Italy (16770, 21416, 22759), Association for International Cancer Research, AICR (ref-14-0338), Telethon-Italy (GGP12171), Centro europeo di Nano-medicina (ref-EP002), Ministero dell'Istruzione, dell'Universita e della Ricerca (MIUR-PRIN- 2015SJLMB9), and the European Commission (ref-316390). G.R.K. is supported by Marie Curie Initial Training Networks (ITN), (FP7 "aDDRess"; Project number: 316390) fellowship and Italian Association for Cancer Research (AIRC) fellowship (Ref. 19464). A.K. is supported by Wellcome Trust/DBT India Alliance intermediate fellowship, SERB (Department of Science and Technology), and Department of Biotechnology, Government of India. J.B. by is supported by the Danish Cancer Society, the Danish Council for Independent Research, the Novo Nordisk Foundation, the Swedish Research Council, and CancerFonden. S.P. receives support from Denovostem ERC 670126 and FARE-MIUR grants. T.P. is a Cariparo-Foundation fellow.

## Author contributions

M.F. and G.R.K. designed all the experiments and wrote the manuscript. G.V.B. and A.M. performed all the EM analysis. G.R.K. and Q.L. performed AFM and compression device measurements. G.B., P.M., D.P., and S.B. contributed towards FRET analysis. Proteomic and lipidomic analysis was done by G.R.K., U.R., V.M., K.H., A.K., and C.B. A.M. and I.G. contributed towards migration assays. E.F. and A.P. performed tail vein injection assays. Z.Z. performed in vitro and in vivo neuronal migration experiments. M.R. and M.P. provided support for micro-fabricated migration assays. S.P. and T.P. provided support for YAP experiments and reagents, and A.K., S.P., G.S., P.M., J.B., Z.Q.W. and A.B. provided constructive inputs and revised the MS.

## Competing interests

The authors declare no competing interests.
