## [Peer Review File · Nature Communications]

Reviewers' comments:

Reviewer #1 (Remarks to the Author):

This paper presents important and interesting new findings relating to the interplay between ATR and communication between the cytoplasm and the nuclear interior (chromatin/chromosomes) as mediated by the nuclear envelope. There are many detailed findings presented that are of interest. Two general points are of primary significance.

First, ATR is traditionally known as a mediator of responses to problems in DNA/chromosomes. However, in seminal previous work, this group has shown that one such response involves topological/mechanical stress within the DNA and occurs via ATR at the nuclear envelope (NE); they also showed that ATR localizes to the NE specifically during S-phase when this response will be required; and that ATR hyperlocalizes to the NE in response to externally applied mechanical force or osmotic pressure. These observations, plus the fact that ATR is a giant HEAT repeat protein, raised the possibility that ATR might directly sense and transduce mechanical stress in general and at the NE in particular.

The current paper now shows that ATR is actually required for the NE itself to have normal properties, thus revealing an entirely new aspect of ATR functionality. Depletion experiments show that ATR has roles for NE morphology in unchallenged cells; but diverse presented studies show that depletion of ATR has especially profound effects in situations where cells (and thus their nuclei, which are their primary load-bearing element) are under mechanical stress by deformations of various types. This is studied in a microfluidic system where cells are forced through narrow channels. Thus, broadly speaking, ATR is a key component of the system by which nuclei (and thus cells) sense and transduce mechanical forces. This is an entirely new discovery and opens the way to many future studies. This work also identifies several suspects for molecules that might be recruiters, stress-transducers and/or targets of ATR.

Second, this paper shows that ATR is essential for two processes in which cells must squeeze through tight spaces (neurogenesis and metastasis), in accord with the above microfluidic studies, and with additional interesting observations regarding the specific problems encountered.

Appreciation of this paper would be significantly improved if it were better written. Multiple specific suggestions are provided below. But the overall issues are (a) to cleanly separate results and interpretation; (b) to reduce overstatements/hype; (c) to better organize both the data and the conclusions. These are challenging tasks because of the large quantity of information and because of the fact that many effects are interrelated. Nonetheless, additional effort in this direction is warranted.

Of special importance is to clarify that (i) what the data show is that ATR is "involved in/required for" many interesting effects but that (ii) the specific role(s) for ATR are unclear. Further (iii) it is attractive to assume that the defects observed imply direct roles of ATR in sensing and transducing mechanical information, for many reasons including economy, HEAT repeats and previous observations in which ATR responds to topological (and thus mechanical) effects of NE-associated replication/transcription collisions. But this is not directly documented. And finally (iv) identified interactions with various molecules could reflect recruitment, mechanosensing and/or transduction of information via ATR-mediated phosphorylation of various targets.

In general the conclusions are very fully documented with appropriate controls. The one issue that does need to be further addressed, or at least clarified, is the assumption that the nesprin FRET sensor is sensing "tension". In fact, it is measuring distance, not tension. It is an inference that distance changes reflect changes in tension. If I understand correctly, changes in distance could result from changes in the molecular lengths of linker molecules, rather than changes in mechanical forces that alter their physical lengths. The cited reference only says that there should be mechanosensing and that nesprin should be involved, but it does not say that nesprin per se is the elastic element. Is there really evidence that the FRET construct is a tension sensor? Are there controls to say this? What happens with AFM perturbation where there is a "real time" response with no chance for molecular reorganization? More experiments are not required at this point - it is only required that the actual situation be clearly explained. If it is correct that it is an "inference" that tension is being sensed, then the results are "consistent" with a reduction in tension when ATR is depleted (rather than "demonstrating" such a reduction).

Also, the authors present one experiment showing that certain defects require ATR-mediated kinase activity. It is not appropriate to include more data in the present paper, for many reasons. But the implication of the presented finding should be stated more clearly (see below) and, in future, it would certainly be interesting to have a better idea of whether all depletion phenotypes, including localization defects, are dependent on the kinase function, which will help to know the specific role(s) of ATR beyond the final downstream phosphorylation.

Suggested writing improvements. Note: the number of comments does not imply any problem with content, only presentation.

1. Title change to: "ATR is essential for preservation of cell mechanics and nuclear integrity during....." The current "preserves" implies direct effects but what is shown is that depletion causes problems.

2. abstract line 23

Is there some way to make the first sentence of the abstract more faithful to what was shown in previous work? The problem is that the previous work does not show that "ATR is activated". It shows that "ATR responds to mechanical stress at the NE and mediates NE-associated repair of aberrant topological DNA states"

3. abstract line 24 starting with "ATR preserves...." This entire sentence can be omitted. Just talk about the data; generalizations are not meaningful in advance of facts.

4. abstract - latter part. I could suggest the following:

"When subjected to mechanical stress or undergoing interstitial migration, ATR-defective nuclei exhibit nuclear collapse, NE ruptures, perinuclear cGAS accumulation, which indicates loss of NE integrity, and aberrant perinuclear chromatin status. ATR-defective cells also are defective in neuronal migration during...(etc). These and other findings indicate that ATR ensures mechanical coupling of the cytoskeleton to the NE and accompanying regulation of NE-chromosome association. Thus: the repertoire of ATR-regulated biological processes extends well beyond its canonical role in triggering biochemical implementation of the DNA damage response."

not perfect, but perhaps the general goal is clear

5. Line 38. What is "nuclear plasticity"? It is important to distinguish between between "plasticity" in the generic sense and the mechanical sense. Some more specific word/wording would be better than this jargon. It may well be understood in the field but is nonetheless ambiguous and there should be a better term.

6. Line 40. Do you mean "nucleus" or "nuclear envelope". Ambiguous in first two sentences. Thus perhaps: "the physical properties of the nucleus are modulated in response to inputs from..." would be better.

7. Line 41. Turn sentence around. "The nuclear envelope (NE) plays a critical role in this process by connecting the cytoskeleton and the chromosomes"

8. Line 50. what do you mean by "nuclear mechanics" More jargon. Do you mean responses of the nucleus to mechanical inputs? Sensing and transducing of mechanical forces by the nucleus? be more clear. The term nuclear mechanics should be eliminated everywhere, in my view.

9. Line 50. "We hypothesize" should be omitted. The reader does not care about hypotheses, only about facts. Plus the preceding list of "considerations" are strung together as a list without clear elucidation of the logic connecting them.

Example fix: Previous observations suggest that ATR at the NE directly senses mechanical stress at the NE/chromatin interface to effect release of chromatin from the NE. This possibility is supported by the fact that ATR comprises heat repeats, which are elastic connectors. Here we explore the possibility that ATR-mediated mechanical communication are also important for the state of the NE itself and, having obtained evidence to this effect, explore its functional implications".

10. The paper would benefit from more frequent subheadings that state observations, not conclusions.

for example:

- line 52 ATR in the cytoplasm localizes preferentially to membranes and actin fillaments and in proximity to the nucleus/NE.
 - line 64 Depletion of ATR results in multiple nuclear membrane defects.
 - line 83 Depletion of ATR alters the response of the nucleus to external force
 - line 92 Depletion of ATR alters diverse molecular features related to nuclear envelope status
- subheadings: lipid composition; heterochromatin formation; inner/outer membrane distance-nesprin length; YAP accumulation

11. line 83: suggestion: Abnormalities in the NE and/or chromatin/chromosomes can affect the mechanical properties of the nucleus

12. line 87: (Fig 2b), consistent with the fact that the nucleus is the stiffest component of the cell (refs). move this statement to this position in the text from farther down.

13. line 87 ff. " The irregular spikes in the force plots (Fig. 2c) in shATR cells suggests that nuclei

collapse due to their inability to sustain the mechanical pressure. The abnormal force plots increased to nearly 50% in shATR cells (Fig. 2c), well correlating with the percentage of deformed nuclei (Fig. 1c)."

The experiment needs to be described more clearly so one doesn't have to look at the figure. Is there a plot of force vs time? over what times? Spikes of what? And separate "in principle" what spikes mean from difference in normal and shATR cells. That is: spikes mean nuclear collapse. See X spikes per time (or probability per cell?) in normal cells and Y% in shATR. And what is the % of nuclear collapse in Figure 1? don't make the reader go look.

14. line 92. New subheading (see above)

15. line 107: ATR does not "cause heterochromatinization"...it "promotes increased heterochromatinization".

16. line 108. say: This conclusion is confirmed by.....

17. Lines 112-114. there are two words missing in these sentences

18. line 115: you say above that there is altered lipid content; now you say it is the chromatin that affects NE behavior. So in this section, you can only say that chromatin state may be another feature that contributes to reduced NE resistance to mechanical force

19. line 117 ff - see above comments re FRET assessment of tension

20. line 141. ATR does not prevent nuclear collapse.....You show that depletion of ATR renders cells hypersensitive to mechanical compression...prevent implies direct role; you don't show a direct role.

21. line 142. You need to say: ATR is a molecule whose presence is required for normal response of the nucleus to mechanical forces. Then, since responses of the cell are dominated by effects on the nucleus, you can generalize to "the cell". Need to make cell/nucleus distinction clearer.

22. line 157. Make this a new section specifically discussing neurogenesis and metastasis.

23. line 159. As above, no one cares about your "reasoning". Just say that many cellular functions involving moving through tight spaces and the observations in the previous section suggest that depletion of ATR should severely compromise the ability of cells to do such moving. To explore this possibility you examined to well-known cases: neurogenesis and metastasis. In the case of neurogenesis, nuclear stresses and accompanying chromatin compaction are prominent features. (There is also information from Discher re metastasis; could be mentioned). Then go on with description of experiments.

24. line 175. tension issue again

25. Line 178 "likely due to their intrinsic defects in nuclear mechanics".. the recurring issue is "what is nuclear mechanics". Be more specific.

26. Line 183: "By analysing the cGAS-GFP foci distribution, we found that in ATR184 defective cells cGAS-GFP foci appeared much earlier than controls at the leading tip of nucleus engaged in the constriction (Fig. 3f)."

This finding should be presented below in the context of other data regarding leading/lagging ends of the nucleus.

27. Line 193 Both control and ATR-depleted cells exhibited sporadic NE ruptures (Fig. 3h), in accordance with previous reports 21,22.

This is ambiguous. Previous papers do NOT talk about ATR, only about control cells. ATR role has not been previously analyzed, yes? Need to make this clear so current data are appreciated.

28. Line 206

While the intrinsic defects in nuclear mechanics in ATR-defective cells may not affect cell viability under normal conditions, the consequences of nuclear collapse following mechanical stress certainly contribute to cell lethality.

"nuclear mechanics"fix

Meaning unclear.....cell lethality when cells are forced through narrow passages, not "cell lethality in general" ????

29. line 211. Omit first sentence. Just say:

Cell migration for neurogenesis and for metastasis involve going through narrow places. Above in vitro suggests ATR important. Consistent with this, neurogenesis involves mechanical stress and previous reports say that state of nuclear membrane is important

30. 225/226. "we predicted" As above.....Rephrase.

31. Line 235 Title: known to influence (better wording) mechanical responses of the nucleus (not the cell)

32. line 246 We identified several ATR interactors able to influence nuclear mechanics and contribute, at least in part, to some of the phenotypes observed in ATR-depleted cells
fix to: We identified several ATR interactors for which previous studies have identified roles in the mechanical properties of the nucleus and whose depletion mimics, at least in part, some of the phenotypes observed in.....

33. line 269-270 It should be clarified that there are various reasons for interaction with ATR: does stress activate ATR phosphorylation which acts on these targets?

And/or do these targets recruit ATR? And/or do these targets participate in ATR activation?

Discussion - in the view of this reviewer, the authors are free to say whatever they wish in the Discussion, but it could be possible to further separate the observations from the conclusions from the inferences while still emphasizing the intrinsic interest of the work. Moreover, the two most important points (see comments at beginning of the review) are somewhat drowned out in the plethora of other information.

Reviewer #2 (Remarks to the Author):

The study by Kidiyoor et al. uses microscopy, biophysical and in-vivo approaches to characterize mechanical stress-associated phenotypes that are observed in ATR deficient cells. The authors show that loss of ATR results in nuclear envelop defects that lead to nuclear collapse under mechanical stress. Such mechanical stress occurs during interstitial cell migration, which the authors show is reduced upon loss of ATR function. The study also shows that loss of ATR impacts a variety of mechanical responses, which includes reducing the nuclear localization of the Hippo pathway component YAP. Further potential insight into ATR function is shown from a mass spectrometry-based screening effort to identify ATR-associated proteins.

The manuscript is well written and clear, and provides interesting details into the functions of ATR. The concept of the study has a somewhat moderate impact given the group's prior study, but the manuscript does offer important information that is interesting to the broader research community. There are a few minor comments that I have, which are listed below:

- The conclusion that H3K9me3 is increased in ATR-depleted cells is weak based on the data shown. The western blot images do not show an obvious change and no statistics are provided for the quantitation. Given how central this conclusion is throughout the manuscript, more solid evidence that chromatin dynamics are changing in ATR deficient cells should be provided.
- No statistics are provided for Figure 3F.
- Figures 4 and 5 are mixed up (based on the text Figure 5 should be Figure 4 and vice versa).
- The images in Figure 5b (referenced as 4b in the text) do not appear to show any obvious differences in cell number. Is this the case? One would expect a reduced cell number give that loss of ATR leads to nuclear collapse and cell death during migration.
- No controls are shown for the PLA experiment in Figure 4C (referenced as 5C in the text). It is unclear whether the signals that are shown are above background.
- While it is nice to see a confirmation of the mass spec with the ATR and Nesprin-2 experiments, it is unclear how this interaction (or others) fit into the mechanism by which ATR regulates nuclear mechanics. The impact of the study would be greatly improved if the authors can connect one of the potential ATR binding partners with the altered mechanical properties observed in ATR-depleted cells, such as the changes in nuclear phospholipid composition shown in Figure 2.

Reviewer #3 (Remarks to the Author):

In this paper, Kidiyoor and colleagues address an interesting question and investigate the consequences of ATR depletion on nuclear morphology and mechanics. There is a growing body of literature indicating that changes in nuclear mechanical properties can impact cell differentiation and migration. Interestingly, the authors observed that ATR may regulate some of these aspects. The approach is straightforward and the paper contains a large amount of work. The work adds to the body of knowledge about the consequence of ATR deficiency, but in the end, one doesn't have a clearer picture of why ATR affects nuclear mechanics and morphology. Much of the current study is essentially descriptive, and constitutes a succession of observations and does not reveal a molecular mechanism to account for the different phenotypes, which is extremely frustrating. Even though some observations are intriguing, the study as it is presented here seems too preliminary to be published in Nature Communications.

-It is unclear whether the distinct observations made in figure 1 and 2 are connected. The authors observed nuclear envelope defects and decreased nuclear stiffness, but is-it a consequence (or a cause?) of the other observations? Such as: the altered nuclear lipid composition (figure 2d)? Or increased chromatin condensation (figure 2e)? Or LINC mediated tension? (Figure 2f which is unlikely, since the AFM experiments have been done using isolated nuclei)

-One of the main question remaining is how does ATR regulate nuclear morphology/structure? They observed that ATR kinase activity was critical in producing these effects. The authors identified many ATR interactors using mass spec (figure 4), however they performed no control experiment to validate these interactions (besides nesprin 2) and they did not test if these proteins were actual ATR substrates (Is nesprin2 phosphorylated by ATR? And if so, what is the consequence of this phosphorylation and can it explain the phenotype?). The authors also show that ATR localize at the membranes (figure 1), suggesting that this could participate to the observed phenotype, but they did not test it (potentially using mutant forms of ATR with or without the putative membrane binding regions).

- AFM data obtained from ATR depleted cells show difference between retraction and approach curve which could be hysteresis, and may be due to viscous components (and not necessarily "nuclei collapse"). The nuclei were isolated using detergent, which alters nuclear membranes and can potentially affect nuclear mechanical properties.

-figure 4b. Nesprin 2 is a very large protein which is hard to observe by western blot, what is the molecular weight of the signal showed in this figure?

- Surprisingly, the authors observed no difference in the number of 53BP1 foci between control cells and cells depleted for ATR (line 167) and conclude that "cell death of ATR defective cells does not correlate [...] with increased DNA damage". This is discordant with previous work from Lammerding's group and Piel's laboratory showing that nuclear envelope rupture leads to DNA damage, do the authors have a potential explanation for this apparent discrepancy?

-The authors observed that YAP activity and phosphorylation is affected by ATR depletion. YAP can be activated in response to NE deformation (Elosegui-Artola et al and Aureille et al.), but can the authors explain how a softer nucleus could be responsible for YAP inhibition (and YAP phosphorylation)?

-It is unclear what the authors mean by "preserve cell mechanics" in the title (and elsewhere in the manuscript)?

Answers to the reviewers' comments.

We thank the reviewers for their suggestions and criticisms. Our answers are in red.

Reviewers' comments:

Reviewer #1 (Remarks to the Author):

This paper presents important and interesting new findings relating to the interplay between ATR and communication between the cytoplasm and the nuclear interior (chromatin/chromosomes) as mediated by the nuclear envelope. There are many detailed findings presented that are of interest. Two general points are of primary significance. First, ATR is traditionally known as a mediator of responses to problems in DNA/chromosomes. However, in seminal previous work, this group has shown that one such response involves topological/mechanical stress within the DNA and occurs via ATR at the nuclear envelope (NE); they also showed that ATR localizes to the NE specifically during S-phase when this response will be required; and that ATR hyperlocalizes to the NE in response to externally applied mechanical force or osmotic pressure. These observations, plus the fact that ATR is a giant HEAT repeat protein, raised the possibility that ATR might directly sense and transduce mechanical stress in general and at the NE in particular. The current paper now shows that ATR is actually required for the NE itself to have normal properties, thus revealing an entirely new aspect of ATR functionality. Depletion experiments show that ATR has roles for NE morphology in unchallenged cells; but diverse presented studies show that depletion of ATR has especially profound effects in situations where cells (and thus their nuclei, which are their primary load-bearing element) are under mechanical stress by deformations of various types. This is studied in a microfluidic system where cells are forced through narrow channels. Thus, broadly speaking, ATR is a key component of the system by which nuclei (and thus cells) sense and transduce mechanical forces. This is an entirely new discovery and opens the way to many future studies. This work also identifies several suspects for molecules that might be recruiters, stress-transducers and/or targets of ATR. Second, this paper shows that ATR is essential for two processes in which cells must squeeze through tight spaces (neurogenesis and metastasis), in accord with the above microfluidic studies, and with additional interesting observations regarding the specific problems encountered.

Appreciation of this paper would be significantly improved if it were better written. Multiple specific suggestions are provided below. But the overall issues are (a) to cleanly separate results and interpretation; (b) to reduce overstatements/hype; (c) to better organize both the data and the conclusions. These are challenging tasks because of the large quantity of information and because of the fact that many effects are interrelated. Nonetheless, additional effort in this direction is warranted.

We thank the referee for the comments and suggestions. The paper has been extensively edited.

Of special importance is to clarify that (i) what the data show is that ATR is "involved in/required for" many interesting effects but that (ii) the specific role(s) for ATR are unclear. Further (iii) it is attractive to assume that the defects observed imply direct roles of ATR in sensing and transducing mechanical information, for many reasons including economy, HEAT repeats and previous observations in which ATR responds to topological (and thus mechanical) effects of NE-associated replication/transcription collisions. But this is not directly documented. And finally (iv) identified interactions with various molecules could reflect recruitment, mechanosensing and/or transduction of information via ATR-mediated phosphorylation of various targets.

The comments are well taken and we therefore described and discussed the results accordingly.

In general the conclusions are very fully documented with appropriate controls. The one issue that does need to be further addressed, or at least clarified, is the assumption that the nesprin FRET sensor is sensing "tension". In fact, it is measuring distance, not tension. It is an inference that distance changes reflect changes in tension. If I understand correctly, changes in distance could result from changes in the molecular lengths of linker molecules, rather than changes in mechanical forces that alter their physical lengths. The cited reference only says that there should be mechanosensing and that nesprin should be involved, but it does not say that nesprin per se is the elastic element. Is there really evidence that the FRET construct is a tension sensor? Are there controls to say this? What happens with AFM perturbation where there is a "real time" response with no chance for molecular reorganization? More experiments are not required at this point - it is only required that the actual situation be clearly explained. If it is correct that it is an "inference" that tension is being sensed, then the results are "consistent" with a reduction in tension when ATR is depleted (rather than "demonstrating" such a reduction).

Agree. We modified the text to tune down the "Tension" issue and to clarify the conclusions.

Also, the authors present one experiment showing that certain defects require ATR-mediated kinase activity. It is not appropriate to include more data in the present paper, for many reasons. But the implication of the presented finding should be stated more clearly (see below) and, in future, it would certainly be interesting to have a better idea of whether all depletion phenotypes, including localization defects, are dependent on the kinase function, which will help to know the specific role(s) of ATR beyond the final downstream phosphorylation.

We added new experiments and changed the text extensively. We can now classify the phenotypes of ATR-defective cells as direct consequences of ATR catalytic inhibition or in long term effects due to chronic ATR depletion.

Suggested writing improvements. Note: the number of comments does not imply any problem with content, only presentation.

1. Title change to: "ATR is essential for preservation of cell mechanics and nuclear integrity during....." The current "preserves" implies direct effects but what is shown is that depletion causes problems.

Done

2. abstract line 23

Is there some way to make the first sentence of the abstract more faithful to what was shown in previous work? The problem is that the previous work does not show that "ATR is activated". It shows that "ATR responds to mechanical stress at the NE and mediates NE-associated repair of aberrant topological DNA states"

Done

3. abstract line 24 starting with "ATR preserves...." This entire sentence can be omitted. Just talk about the data; generalizations are not meaningful in advance of facts.

Done

4. abstract - latter part. I could suggest the following:

"When subjected to mechanical stress or undergoing interstitial migration, ATR-defective nuclei exhibit nuclear collapse, NE ruptures, perinuclear cGAS accumulation, which indicates loss of NE integrity, and aberrant perinuclear chromatin status. ATR-defective cells also are defective in neuronal migration during...(etc). These and other findings indicate that ATR ensures mechanical coupling of the cytoskeleton to the NE and accompanying regulation of NE-chromosome association. Thus: the repertoire of ATR-regulated biological processes extends well beyond its canonical role in triggering biochemical implementation of the DNA damage response."

not perfect, but perhaps the general goal is clear

Done

5. Line 38. What is "nuclear plasticity"? It is important to distinguish between "plasticity" in the generic sense and the mechanical sense. Some more specific word/wording would be better than this jargon. It may well be understood in the field but is nonetheless ambiguous and there should be a better term.

Changed

6. Line 40. Do you mean "nucleus" or "nuclear envelope". Ambiguous in first two sentences. Thus perhaps: "the physical properties of the nucleus are modulated in response to inputs from..." would be better.

The sentence has been reorganized as suggested.

7. Line 41. Turn sentence around. "The nuclear envelope (NE) plays a critical role in this process by connecting the cytoskeleton and the chromosomes"

Done

8. Line 50. what do you mean by "nuclear mechanics" More jargon. Do you mean responses of the nucleus to mechanical inputs? Sensing and transducing of mechanical forces by the nucleus? be more clear. The term nuclear mechanics should be eliminated everywhere, in my view.

The paragraph has been reorganized as suggested.

9. Line 50. "We hypothesize" should be omitted. The reader does not care about hypotheses, only about facts. Plus the preceding list of "considerations" are strung together as a list without clear elucidation of the logic connecting them. Example fix: Previous observations suggest that ATR at the NE directly senses mechanical stress at the NE/chromatin interface to effect release of chromatin from the NE. This possibility is supported by the fact that ATR comprises heat repeats, which are elastic connectors. Here we explore the possibility that ATR-mediated mechanical communication are also important for the state of the NE itself and, having obtained evidence to this effect, explore its functional implications".

Done (much better now!)

10. The paper would benefit from more frequent subheadings that state observations, not conclusions.

for example:

- line 52 ATR in the cytoplasm localizes preferentially to membranes and actin filaments and in proximity to the nucleus/NE.

- line 64 Depletion of ATR results in multiple nuclear membrane defects.

- line 83 Depletion of ATR alters the response of the nucleus to external force

- line 92 Depletion of ATR alters diverse molecular features related to nuclear envelope status

subheadings: lipid composition; heterochromatin formation; inner/outer membrane distance-nesprin length; YAP accumulation

Done

11. line 83: suggestion: Abnormalities in the NE and/or chromatin/chromosomes can affect the mechanical properties of the nucleus

Done

12. line 87: (Fig 2b), consistent with the fact that the nucleus is the stiffest component of the cell (refs). move this statement to this position in the text from farther down.

Done

13. line 87 ff. " The irregular spikes in the force plots (Fig. 2c) in shATR cells suggests that nuclei collapse due to their inability to sustain the mechanical pressure. The abnormal force plots increased to nearly 50% in shATR cells (Fig. 2c), well correlating with the percentage of deformed nuclei (Fig. 1c)." The experiment needs to be described more clearly so one doesn't have to look at the figure. Is there a plot of force vs time? over what times? Spikes of what? And separate "in principle" what spikes mean from difference in normal and shATR cells. That is: spikes mean nuclear collapse. See X spikes per time (or probability per cell?) in normal cells and Y% in shATR. And what is the % of nuclear collapse in Figure 1? don't make the reader go look.

We removed the curves from figure and modified the text accordingly.

14. line 92. New subheading (see above)

Done

15. line 107: ATR does not "cause heterochromatinization" ...it "promotes increased heterochromatinization".

Done

16. line 108. say: This conclusion is confirmed by.....

Done

17. Lines 112-114. there are two words missing in these sentences

We changed the sentence

18. line 115: you say above that there is altered lipid content; now you say it is the chromatin that affects NE behavior. So in this section, you can only say that chromatin state may be another feature that contributes to reduced NE resistance to mechanical force

This section has been extensively edited

19. line 117 ff - see above comments re FRET assessment of tension

Done. We changed the text extensively to clarify the conclusions of the experiment.

20. line 141. ATR does not prevent nuclear collapse.....You show that depletion of ATR renders cells hypersensitive to mechanical compression...prevent implies direct role; you don't show a direct role.

Agree and changed

21. line 142. You need to say: ATR is a molecule whose presence is required for normal response of the nucleus to mechanical forces. Then, since responses of the cell are dominated by effects on the nucleus, you can generalize to "the cell". Need to make cell/nucleus distinction clearer.

Done

22. line 157. Make this a new section specifically discussing neurogenesis and metastasis.

Done

23. line 159. As above, no one cares about your "reasoning". Just say that many cellular functions involving moving through tight spaces and the observations in the previous section suggest that depletion of ATR should severely compromise the ability of cells to do such moving. To explore this possibility you examined to well-known cases: neurogenesis and metastasis. In the case of neurogenesis, nuclear stresses and accompanying chromatin compaction are prominent features. (There is also information from Discher re metastasis; could be mentioned). Then go on with description of experiments.

Done

24. line 175. tension issue again

Done

25. Line 178 "likely due to their intrinsic defects in nuclear mechanics".. the recurring issue is "what is nuclear mechanics". Be more specific.

The sentence has been changed

26. Line 183: "By analysing the cGAS-GFP foci distribution, we found that in ATR184 defective cells cGAS-GFP foci appeared much earlier than controls at the leading tip of nucleus engaged in the constriction (Fig. 3f)."

This finding should be presented below in the context of other data regarding leading/lagging ends of the nucleus.

Done

27. Line 193 Both control and ATR-depleted cells exhibited sporadic NE ruptures (Fig. 3h), in accordance with previous reports 21,22.

This is ambiguous. Previous papers do NOT talk about ATR, only about control cells. ATR role has not been previously analyzed, yes? Need to make this clear so current data are appreciated.

Done

28. Line 206

While the intrinsic defects in nuclear mechanics in ATR-defective cells may not affect cell viability under normal conditions, the consequences of nuclear collapse following mechanical stress certainly contribute to cell lethality. "nuclear mechanics"....fix

Meaning unclear.....cell lethality when cells are forced through narrow passages, not "cell lethality in general" ????

Fixed

29. line 211. Omit first sentence. Just say:

Cell migration for neurogenesis and for metastastasis involve going through narrow places. Above in vitro suggests ATR important. Consistent with this, neurogenesis involves mechanical stress and previous reports say that state of nuclear membrane is important

Done, see also the previous paragraph.

30. 225/226. "we predicted" As above.....Rephrase.

Done

31. Line 235 Title: known to influence (better wording) mechanical responses of the nucleus (not the cell)

Done

32. line 246 We identified several ATR interactors able to influence nuclear mechanics and contribute, at least in part, to some of the phenotypes observed in ATR-depleted cells

fix to: We identified several ATR interactors for which previous studies have identified roles in the mechanical properties of the nucleus and whose depletion mimics, at least in part, some of the phenotypes observevd in.....

Done

33. line 269-270 It should be clarified that there are various reasons for interaction with ATR: does stress activate ATR phosphorylation which acts on these targets?

And/or do these targets recruit ATR? And/or do these targets participate in ATR activation?

Done

Discussion - in the view of this reviewer, the authors are free to say whatever they wish in the Discussion, but it could be possible to further separate the observations from the conclusions from the inferences while still emphasizing the intrinsic interest of the work. Moreover, the two most important points (see comments at beginning of the review) are somewhat drowned out in the plethora of other information.

Agree. The discussion has been extensively modified.

Reviewer #2 (Remarks to the Author):

The study by Kidiyoor et al. uses microscopy, biophysical and in-vivo approaches to characterize mechanical stress-associated phenotypes that are observed in ATR deficient cells. The authors show that loss of ATR results in nuclear envelop defects that lead to nuclear collapse under mechanical stress. Such mechanical stress occurs during interstitial cell migration, which the authors show is reduced upon loss of ATR function. The study also shows that loss of ATR impacts a variety of mechanical responses, which includes reducing the nuclear localization of the Hippo pathway component YAP. Further potential insight into ATR function is shown from a mass spectrometry-based screening effort to identify ATR-associated proteins.

The manuscript is well written and clear, and provides interesting details into the functions of ATR. The concept of the study has a somewhat moderate impact given the group's prior study, but the manuscript does offer important information that is interesting to the broader research community. There are a few minor comments that I have, which are listed below:

We thank the referee for the comments and suggestions. The paper has been extensively edited.

- The conclusion that H3K9me3 is increased in ATR-depleted cells is weak based on the data shown. The western blot images do not show an obvious change and no statistics are provided for the quantitation. Given how central this conclusion is throughout the manuscript, more solid evidence that chromatin dynamics are changing in ATR deficient cells should be provided.

We have now included the statistics (see Supplementary Figure 2d). Moreover, we have performed new sets of experiments. The DNase sensitivity assay (Supplementary Figure 2c) is routinely used to assess the chromatin state and is based on the rationale that DNase preferentially cleaves euchromatin. We show that ATR defective cells at early time points are more resistant to DNase treatment than control cells. We conclude that ATR defective cells exhibit a lower euchromatin/heterochromatin ratio compared to control. Moreover, we compared the accumulation of H3K9-trimethylation and chromatin compaction by FLIM-FRET following acute treatments with ATR inhibitors or after chronic ATR depletion; we found that both phenotypes represent long term responses following ATR depletion. Notably, a recent paper (Nava et al. 2020) showed that H3K9-3Me heterochromatic levels rearrange in response to mechanical stress at the NE and when nuclei recover from the stress. Our new results suggest that the accumulation of H3K9-trimethylation may reflect the inability of ATR depleted cells to recover from nuclear stress, rather than representing a direct consequence of ATR inactivation. Most likely, the aberrant nuclear morphology of ATR defective nuclei, which is also a late effect, does not allow the cells to re-form a correct perinuclear NE-heterochromatin organization. In these months we also tried to map the heterochromatic regions at the genomic levels using antibodies against H3K9-trimethylation, but we experienced some technical difficulties that could not be solved also due to the covid crisis that imposed a lock down in the institute. Anyhow, we hope that the new results and the text editing addresses the reviewer's comments.

- No statistics are provided for Figure 3F.

We have now included statistics

- Figures 4 and 5 are mixed up (based on the text Figure 5 should be Figure 4 and vice versa).

Sorry about that. We now corrected the figures

- The images in Figure 5b (referenced as 4b in the text)

Corrected

do not appear to show any obvious differences in cell number. Is this the case? One would expect a reduced cell number given that loss of ATR leads to nuclear collapse and cell death during migration.

We note that the cells may not die since they exit cell cycle and stop migrating at E18.5 (4 days after electroporation). In fact, it is difficult to make any conclusion on cell death from the number of quantified cells since some sections have more cells whereas others have less. As you can see from the images, shATR-6 treated sections have less cells compared to shLuc or shATR-4.

- No controls are shown for the PLA experiment in Figure 4C (referenced as 5C in the text).

Corrected. Now the controls are shown in Supplementary Figure-4h.

It is unclear whether the signals that are shown are above background.

In supplementary Figure-4h we show that non-specific background signals appear more spread in the cytoplasm, while the specific foci localize in the proximity of the NE.

- While it is nice to see a confirmation of the mass spec with the ATR and Nesprin-2 experiments, it is unclear how this interaction (or others) fit into the mechanism by which ATR regulates nuclear mechanics. The impact of the study would be greatly improved if the authors can connect one of the potential ATR binding partners with the altered mechanical properties observed in ATR-depleted cells, such as the changes in nuclear phospholipid composition shown in Figure 2.

The new sets of results strongly suggest that following an acute treatment with ATR inhibitors cells experience a mechanical uncoupling of cytoskeleton and NE and accumulate YAP in the cytoplasm. The other phenotypes become obvious much later, when cells experience chronic ATR depletion. In this scenario, the Nesprin-2 results become particularly relevant as it is directly implicated in the coupling between NE and cytoskeleton. However, we wish to point out that the ATR-mediated control of the mechanical properties of the nucleus is not a linear pathway (see below). The

proteome data strongly suggest that the phenotypes of ATR defective cells are likely caused by the deregulation of several processes/pathways (see also the scheme presented in the model figure) and some of the factors identified are well known ATR targets. Moreover, we wish to stress that the ATR interactors identified in this study might be involved in the cellular response to mechanical stress by mediating ATR recruitment at the NE, transducing the mechanical stress signals or in triggering short and long term response pathways. Clearly, the characterization of each interaction at a biochemical and genetic level goes beyond the scope of the current work. Our published and unpublished results are in line with our conclusions: 1) Yeast Mlp1 nucleoporin is regulated by ATR/Mec1 and plays a key role in controlling mRNA export and influences chromatin-NE dynamics (Bermejo et al. Cell, 2011); We found that TPR, the human homologue of Mlp1, also in mammals is phosphorylated by ATR and controls mRNA export through the NE; However, TPR contributes only in part to the NE abnormalities observed in ATR defective cells (Kosar and Foiani submitted). For instance, the association of NE with nucleoli does not depend on TPR. 2) We know that Chk1, the primary kinase downstream of ATR, responds to nuclear stress (Kumar et al.2014) and prevents the accumulation of Nuclear Envelope invaginations and YAP de-localization (like ATR); however, Chk1 does not influence the detachment of chromatin and nucleoli from the NE (see Figure 1 for reviewers only). Moreover, Chk1 itself interacts with several proteins which play a role at the NE (Blasius et.al 2011) (see Table 1 for reviewers only), such Lamin A/C, TPR, NUP153 and several importin subunits. Hence, CHK1 defective cells exhibit only part of the nuclear abnormalities of ATR defective cells, implying that the NE abnormalities of ATR defective cells results from the inability of ATR to detach chromatin from the envelope and from the inability of Chk1 to prevent NE invaginations.

Regarding the changes in phospholipid composition of NE membranes. We performed additional experiments to address whether alterations in membrane phospholipid composition could phenocopy some of the nuclear abnormalities of ATR-defective cells. As shown in Figure 2 for reviewers only, we treated cells with Bromoenol Lactone (BEL), that inhibits the calcium-independent Phospholipase A2 (iPLA2), leading to the accumulation of high levels of polyunsaturated PC species (Zhang et al., 2007); We found that BEL treatment reduced the nuclear stiffness of control nuclei to the levels of shATR nuclei, whilst it had a modest effect on shATR nuclei (Figure 2 for reviewers only). However, we feel uncomfortable to include these results as, at the moment, it is not clear whether the changes in phospholipid compositions reflect an ATR-mediated response to mechanical stress or an adaptative metabolic rewiring due to extensive NE remodeling. Interestingly, mechanical stress has been shown to induce metabolic rewiring (Park et al 2020). Moreover, we did not find any ATR interactor involved in phospholipid metabolism.

Reviewer #3 (Remarks to the Author):

In this paper, Kidiyoor and colleagues address an interesting question and investigate the consequences of ATR depletion on nuclear morphology and mechanics. There is a growing body of literature indicating that changes in nuclear mechanical properties can impact cell differentiation and migration. Interestingly, the authors observed that ATR may regulate some of these aspects. The approach is straightforward and the paper contains a large amount of work. The work adds to the body of knowledge about the consequence of ATR deficiency, but in the end, one doesn't have a clearer picture of why ATR affects nuclear mechanics and morphology. Much of the current study is essentially descriptive, and constitutes a succession of observations and does not reveal a molecular mechanism to account for the different phenotypes, which is extremely frustrating. Even though some observations are intriguing, the study as it is presented here seems too preliminary to be published in Nature Communications.

Perhaps in the original version of the paper we did not stress enough the relevance of our findings. To our view, the following findings are of pivotal relevance: 1) ATR influence the mechanical coupling between NE and the cytoskeleton (see also below and the new observations), and 2) the pathological consequences of ATR inactivation/depletion which impact on interstitial migration, neurogenesis and even metastasis.

ATR is known as a key DNA damage response regulator that senses ssDNA to protect genome integrity. Our observations clearly extend ATR functions well beyond its role in sensing single stranded DNA. In particular, while our findings can explain the interstitial migration defects of ATR-depleted cells and the progeroid features and the stem cell exhaustion of Seckel patients and ATR conditional k.o.mice (see our scheme/model in the paper), the canonical function of ATR cannot justify these abnormalities. In fact, ATR defects are reminiscent of those features caused by laminopathies and our recent unpublished observations (Giovannetti, Kidiyoor and Foiani, manuscript in preparation submission) showing that Seckel patients develop cardiomyopathies due to NE anomalies further reinforce our conclusions. We hope that the referee will find the new version of the paper suitable for Nat. Communications.

-It is unclear whether the distinct observations made in figure 1 and 2 are connected. The authors observed nuclear envelope defects and decreased nuclear stiffness, but is-it a consequence (or a cause?) of the other observations? Such as: the altered nuclear lipid composition (figure 2d)? Or increased chromatin condensation (figure 2e)? Or LINC mediated tension? (Figure 2f which is unlikely, since the AFM experiments have been done using isolated nuclei)

The paper has been extensively edited also to clarify this specific point. In general, the phenotypes observed in ATR depleted/inhibited cells can be classified in short term effects (3-4 hours after ATR inhibition) and long term effects (2-7 days following ATR depletion). While some of these phenotypes may represent direct consequences of ATR inactivation others may result from long term adaptive responses. This is now extensively discussed in the new version of the paper. Below we provide a table showing which phenotypes can be observed 3-4 hours after ATR inhibition (short term) or 3-7 days following ATR depletion (long term) and discuss the observed phenotypes one by one.

Lipid changes: *Altogether our observations suggest that the lipid changes at the level of nuclear membranes represent a long term effect, likely due to the extensive nuclear envelope remodeling as also shown in Figure 5h. This is also consistent with the finding that ATR is essential for NE reformation/repair during mitosis (see also Kumar et al. 2014) or following mechanical stress. A likely possibility is that in ATR defective cells NE reformation/repair is inefficient and undergoes extensive remodeling, also by engaging other cellular membranes. This is in agreement with the observations that we did not observe metabolic imbalance at the level of lipid metabolism in ATR-defective cells compared to control, and we did not identify ATR interactors/targets involved in lipid metabolism. Moreover, studies performed in yeast mutants defective in Mec1^{ATR} further demonstrate that ATR does not contribute to control cell metabolism; rather, metabolic imbalances can affect the viability of mec1 mutants through processes mediated by PP2A (Ferrari et al. Mol Cell 2017) and by histone levels (Bruhn et al Nat. Comm. Submitted).*

Nuclear morphology: *Nuclear morphological defects become obvious after chronic ATR depletion, while they are not visible following an acute treatment with ATR inhibitors. We already knew from previous work (Bermejo et al 2011) that ATR controls mRNA export by phosphorylating key nucleoporins and our screen identified several proteins involved in ribosomal RNA export. Moreover, another lab showed that ATR also controls tRNA export (Ghavidel, A. et al 2007). We strongly believe that deregulated RNA export in ATR defective cells can contribute, in the long term, to NE deformations (this is obvious in the case of nucleolar canals). Moreover, the inefficient coordination between chromatin condensation and NE break down in cycling cells exposed to acute ATR-inhibition (Kumar et al. 2014) can also account for the long term accumulation of nuclear morphological defects. See the final scheme/model in the paper.*

Chromatin alteration: *The new findings show that the changes at the level of chromatin state in ATR-defective cells represent long term effects. A recent paper (Nava et al. 2020) showed that H3K9-3Me heterochromatic levels rearrange in response to mechanical stress at the NE and when nuclei recover from the stress. Our new results suggest that the accumulation of H3K9-trimethylation may reflect the inability of ATR defective cells to recover from nuclear stress, rather than representing a direct consequence of ATR inactivation. Most likely, the aberrant nuclear morphology of ATR defective nuclei, which is also a late effect, does not allow the cells to re-form a correct perinuclear NE-heterochromatin organization.*

NE-cytoskeleton connections and YAP localization: *Our new data indicate that both the mechanical uncoupling between NE and cytoskeleton and YAP delocalization occur immediately after the acute treatments with ATR inhibitors. In ATR-inhibited cells the changes in the Nesprin-2-FRET sensor represent a short term phenotype which persists at long term following ATR depletion. The same is true for YAP delocalization which, based on the literature is an immediate consequence of defective NE-cytoskeleton connections (Elosegui-Artola et al). Our findings that Nesprin-2*

and ATR interact and colocalize and, previous data indicating that the LINC complex is targeted by ATR (Matsuoka et al.,2007), further re-inforce the conclusion that the loose NE-cytoskeleton connections is a direct consequence of ATR inhibition.

Stiffness: the loss of cell/nuclear stiffness occurs after few days following ATR depletion and cannot be observed at short term after acute ATR inhibition (Supplementary fig 2a). These observations suggest that the changes in stiffness likely result from other defects, such as altered lipid composition, aberrant nuclear morphology and chromatin state. Accordingly, we now show that the BEL inhibitor altering the phospholipid metabolism, causes a decrease in stiffness, phenocopying ATR defects (Figure 2 for reviewers only).

Phenotype	Long term	Short term
Abberent nuclear morphology	Yes	No
Loss of nuclear Stiffness	Yes	No (Supplementary fig 2a)
Altered Chromatin state	Yes	No
NE/Cytoskeleton uncoupling	Yes	Yes
YAP cytoplasmic retention	Yes	Yes
Altered PC/PE ratio	Yes	N/A

-One of the main question remaining is how does ATR regulate nuclear morphology/structure? They observed that ATR kinase activity was critical in producing these effects. The authors identified many ATR interactors using mass spec (figure 4), however they performed no control experiment to validate these interactions (besides nesprin 2) and they did not test if these proteins were actual ATR substrates (Is nesprin2 phosphorylated by ATR? And if so, what is the consequence of this phosphorylation and can it explain the phenotype?).

The new sets of results strongly suggest that following an acute treatment with ATR inhibitors cells experience a mechanical uncoupling of cytoskeleton and NE and accumulate YAP in the cytoplasm. The other phenotypes become obvious much later, when cells experience chronic ATR depletion. In this scenario, the Nesprin-2 results become particularly relevant as it is directly implicated in the coupling between NE and cytoskeleton. However, we wish to point out that the ATR-mediated control of the mechanical properties of the nucleus is not a linear pathway (see below). The proteome data strongly suggest that the phenotypes of ATR defective cells are likely caused by the deregulation of several processes/pathways (see also the scheme presented in the model figure) and some of the factors identified are well known ATR targets. Moreover, we wish to stress that the ATR interactors identified in this study might be involved in the cellular response to mechanical stress by mediating ATR recruitment at the NE, transducing the mechanical stress signals or in triggering short and long term response pathways. Clearly, the characterization of each interaction at a biochemical and genetic level goes beyond the scope of the current work. Our published and unpublished results are in line with our conclusions: 1) Yeast Mlp1 nucleoporin is regulated by ATR/Mec1 and plays a key role in controlling mRNA export and influences chromatin-NE dynamics (Bermejo et al. Cell, 2011); We found that TPR, the human homologue of Mlp1, also in mammals is phosphorylated by ATR and controls mRNA export through the NE; However, TPR contributes only in part to the NE abnormalities observed in ATR defective cells (Kosar and Foiani submitted). For instance, the association of NE with nucleoli does not depend on TPR. 2) We know that Chk1, the primary kinase downstream of ATR, responds to nuclear stress (Kumar et al.2014) and prevents the accumulation of Nuclear Envelope invaginations and YAP de-localization (like ATR); however, Chk1 does not influence the detachment of chromatin and nucleoli from the NE (see Figure 1 for reviewers only). Moreover, Chk1 itself interacts with several proteins which play a role at the NE (Blasius et.al 2011) (see Table 1 for reviewers only), such Lamin A/C, TPR, NUP153 and several importin subunits. Hence, CHK1 defective cells exhibit only part of the nuclear abnormalities of ATR defective cells, implying that the NE abnormalities of ATR defective cells results from the inability of ATR to detach chromatin from the envelope and from the inability of Chk1 to prevent NE invaginations.

Finally, please also note that Nesprin 2 defects phenocopies some of the abnormalities, typical of ATR altered cells, (loss of nuclear stiffness and NE invaginations), but not the aberrant association between NE and chromatin or nucleoli. Nesprin 2 contains more than 40 potential phopshorylation sites for ATR (which, by itself it is very exciting!). Clearly we cannot mutagenize 40 sites and, at the best, we could never phenocopy the entire sets of ATR defects as stated above.

The authors also show that ATR localize at the membranes (figure 1), suggesting that this could participate to the observed phenotype, but they did not test it (potentially using mutant forms of ATR with or without the putative membrane binding regions).

We are currently characterizing the ATR domains involved in the mechanical response using hypotonic and hypertonic conditions. Our preliminary observations indicate that the osmotic stress response element of ATR spans between 600 and 1700 AA. However, this is an ongoing work, complicated by the fact that ATR localizes both at the NE as well as at nucleoli following osmotic stress (Kumar et al. 2014) and we are still trying to uncouple the two localization. Furthermore, We now know that ATR defects contribute to heart inflammation and cardiomyopathies, typical defects observed in certain laminopathies (our unpublished observations). These findings besides, supporting our conclusions that ATR defects cause pathological outcomes due to NE defects, allowed us to identify specific ATR mutations that are present as germline mutations in patients developing cardiomyopathies. These allele specific ATR mutations do not affect the DNA damage response and might allow us to uncouple the ATR regulatory processes, to characterize the mechanoresponsive ATR domains, and to identify key players in the ATR-mediated mechano-response. All these data will be part of another story.

- AFM data obtained from ATR depleted cells show difference between retraction and approach curve which could be hysteresis, and may be due to viscous components (and not necessarily "nuclei collapse"). The nuclei were isolated using detergent, which alters nuclear membranes and can potentially affect nuclear mechanical properties.

We removed the curves from figure and modified the text accordingly.

-figure 4b. Nesprin 2 is a very large protein which is hard to observe by western blot, what is the molecular weight of the signal showed in this figure?

It is now showed, (supplementary figure 4h)

- Surprisingly, the authors observed no difference in the number of 53BP1 foci between control cells and cells depleted for ATR (line 167) and conclude that "cell death of ATR defective cells does not correlate [...] with increased DNA damage". This is discordant with previous work from Lammerding's group and Piel's laboratory showing that nuclear envelope rupture leads to DNA damage, do the authors have a potential explanation for this apparent discrepancy?

We analysed the 53BP1 foci in ATR-inhibited cells undergoing interstitial migration and found comparable foci numbers to control (Figure 4c). This finding implies that the DNA damage foci accumulating during interstitial migration do not require ATR for repair. In fact, Raab et al and Denais et al., reported that the DNA damages accumulated by cells migrating into pores resemble double strand breaks that require ATM function. Please note that in ATR defective cells ATM is fully functional. In other words, ATR depleted cells resemble LMNA depleted cells reported in Raab et al., instead of behaving like ATM inhibited cells. We also show that cell lethality induced by interstitial migration in ATR defective cells is uncoupled from the number of 53BP1 foci: control cells exhibiting a high number of foci survive the migration and ATR-defective cells with low number of foci die during the migration in the constrictions.

-The authors observed that YAP activity and phosphorylation is affected by ATR depletion. YAP can be activated in response to NE deformation (Elosegui-Artola et al and Aureille et al.), but can the authors explain how a softer nucleus could be responsible for YAP inhibition (and YAP phosphorylation)?

While our data show that YAP localization and NE-cytoskeleton connections are direct consequences of ATR deregulation (acute inhibition shows the defect) the softer nucleus phenotype becomes obvious after long-term depletion of ATR, most likely due to alterations in nuclear membrane composition and chromatin architecture. Our results are consistent with the report by Elosegui-Artola et al. which also shows that cells that have lower Nesprin 2 mediated NE tension, accumulate cytoplasmic YAP.

-It is unclear what the authors mean by "preserve cell mechanics" in the title (and elsewhere in the manuscript)?

We intend that ATR influences the mechanical properties of the nucleus and of the cell (also due to YAP cytoplasmic accumulation). Due to space limitations we kept cell mechanics in the title but we clarified it in the text. In any case in the new version we better clarified the roles of ATR.

Reviewers' only graphic materials

Figure 1: Nuclear abnormalities in Chk1 Defective cells.

A) EM images of CHK1 (siRNA) depleted HeLa cells. B) AFM measurement of elastic modulus of cells treated with DMSO or CHK1 inhibitor (UCN01 300nM). C. Confocal images of YAP staining in cells treated with DMSO or CHK1 inhibitor (AZD7762 - 50nM-4hr). D) Quantification of nuclear to cytoplasmic ratio.

Figure 2: Nuclear Stiffness can be altered by altering membrane lipid composition

AFM based measurement of isolated nuclear stiffness from control and shATR nuclei untreated or overnight treated with BEL for 16 hours (12μM).

Table 1: Relevant ATR and Chk1 targets

Protein	Reference	Interaction	ATR phosph consensus	Chk1 Phosphorylation
NUP 160	This work	Physical	No	No
NUP 133	This work	Physical	No	No
NUP107	[1], this work	Physical and Phosphorylation	Yes	No
NUP50	[2], this work	Physical and Phosphorylation	Yes	No
TPR	[1,3-4]	Genetic and Phosphorylation	Yes	Yes

TOP2A/B	[1-2,5], this work	Genetic, physical and Phosphorylation	Yes	No
NESPRIN1/2	[2], this work	Physical and Phosphorylation	Yes	No
SUN1/2	[2,6]	Physical and Phosphorylation	Yes	No
CHD4	[1,7], this work	Physical and Phosphorylation	Yes	No
HDAC2	[7-8], this work	Physical	No	No
ESCRT III subunits	[2][9],	Physical and Funcrional	Yes	No

- [1] S Matsuoka et al., *Science* **316** (5828), 1160 (2007).
[2] JV Olsen et al., *Sci Signal* **3** (104), ra3 (2010).
[3] M Blasius et al., *Genome Biol* **12** (8), R78 (2011).
[4] R Bermejo et al., *Cell* **146** (2), 233 (2011).
[5] N Hashash et al., *PLoS Genet* **8** (10), e1002978 (2012).
[6] MP Stokes et al., *Proc Natl Acad Sci U S A* **104** (50), 19855 (2007).
[7] DR Schmidt and SL Schreiber, *Biochemistry* **38** (44), 14711 (1999).
[8] T Robert et al., *Nature* **471** (7336), 74 (2011).
[9] C C Okasa et al., Manucript in communication (2020).

REVIEWERS' COMMENTS

Reviewer #1 (Remarks to the Author):

The authors have addressed all of the concerns that I raised in my original review. Thus, in my opinion, this paper is now suitable for publication in Nature Communications. I may also comment on the concerns of Reviewer 3, who felt that the presented observations were too descriptive (rather than mechanistic).

In my opinion the authors have appropriately addressed the remaining comments of Reviewer 3, and I deem the manuscript suitable for publication.

Here is a more detailed rationale for the two above opinions.

General significance. From ATR was first discovered in 1988 (as Mec1 in yeast) until 5 years ago, this centrally important molecule was thought simply to be a key DNA-based signal transduction molecule which recognized single-stranded DNA, often but not always in the context of DNA damage, and initiated a cascade of phosphorylations through which the cell responded in appropriate ways. Five years ago, the Foiani lab demonstrated that ATR mediated its effects by sensing mechanical stress at the DNA/chromatin - nuclear envelope interface. This groundbreaking finding was supported by the fact that ATR is a giant HEAT repeat protein and that such proteins are known to sense and transduce mechanical stress. Nonetheless, the known activities of ATR still pertained only to the chromosomes. This new paper from the Foiani laboratory now shows that ATR also serves as a mechano-sensitive signal transduction protein for the cytoskeleton/nuclear envelope interface. This is an entirely new universe of activities for this molecule (which previously was known only to be involved in chromosomal events). Furthermore, this paper documents a number of pathological consequences that evolve (directly or indirectly) from depletion or inactivation of ATR including interstitial migration, neurogenesis, metastasis and several other conditions. And finally, this paper specifically implicates Nesperin2 as a key downstream target. These findings make this paper very suitable for publication in Nature Communications.

Reviewer 3 remarks.

First: Reviewer 3's original main complaint was that this paper did not discuss the molecular mechanisms of the observed effects. In my opinion, this is not a valid criticism. This paper is groundbreaking. It opens up many new areas for further investigation. Understanding the detailed mechanisms is a life's work.

Second: The authors have extensively, carefully and exhaustively addressed all of Reviewer 3's concerns. Of primary importance is that they now make a much clearer distinction between immediate effects of ATR depletion/inhibition (which define direct roles) and longer-term effects (which define secondary downstream consequences). A second issue pertains to the proteomics analysis, which the reviewer found too diffuse. The authors have now made clear that this analysis specifically supports identification of Nesprin 2 as a primary target, that many other identified molecules are known targets of ATR, and that the results can be integrated into a multi-pathway working model. Overall, I did not find any Reviewer 3 concern that was not satisfactorily addressed.

Summary. This is a very important paper which fully merits publication in Nature Communications.

Reviewer #2 (Remarks to the Author):

The revised manuscript has been much improved and addresses most initial comments. In my opinion the manuscript is suitable for publication.